# Using COVID-19 pandemic perturbation to model RSV-hMPV interactions and potential implications under RSV interventions

Emily Howerton [1] ✉, Thomas C. Williams [2,3], Jean-Sébastien Casalegno [4], Samuel Dominguez[5], Rory Gunson[6], Kevin Messacar [5], C. Jessica E. Metcalf[1], Sang Woo Park [1,7], Cécile Viboud [8] & Bryan T. Grenfell [1]

Respiratory syncytial virus (RSV) and human metapneumovirus (hMPV) are closely related pathogens responsible for a significant burden of acute respiratory infections. Interactions between RSV and hMPV have been hypothesized, but the mechanisms of interaction are largely unknown. Here, we use a mathematical model to quantify the likelihood of interactions from population-level surveillance data and investigate whether interactions could lead to increases in hMPV burden under RSV medical interventions, including active and passive immunization. In Scotland, Korea, and three regions of Canada, annual hMPV outbreaks lag RSV outbreaks by up to 18 weeks; two Canadian regions show patterns consistent with out-of-phase biennial outbreaks. Using a two-pathogen transmission model, we show that a negative effect of RSV infection on hMPV transmissibility can explain these dynamics. We use post-pandemic RSV-hMPV rebound dynamics as an out of sample test for our model, and the model with interactions better predicts this period than a model where the pathogens are assumed to be independent. Finally, our model suggests that hMPV peak timing and magnitude may change under RSV interventions. Our analysis provides a foundation for detecting possible RSV-hMPV interactions at the population level, although such a model over-simplifies important complexities about interaction mechanisms.

Respiratory syncytial virus (RSV) and human metapneumovirus (hMPV) are closely related pneumoviruses that cause a significant burden of acute respiratory infection in young children[1–4]. Beyond close genetic relatedness, RSV and hMPV share many key features, including a number of similar sites on the fusion protein[5], propensity to infect epithelial cells in the respiratory tract[6], and indistinguishable clinical presentation[7].

RSV and hMPV outbreaks are also characterized by clear dynamical patterns. Across the globe, RSV typically exhibits annual or biennial winter outbreaks, which are driven in part by climatic factors[8,9]. In temperate regions, hMPV outbreaks usually lag RSV outbreaks by a few weeks or longer[2], although the mechanism that underlies this pattern is unknown. One null hypothesis is that RSV and hMPV circulate independently and cause outbreaks at different times

[1]Department of Ecology and Evolutionary Biology, Princeton University, Princeton, NJ, USA. [2]Child Life and Health, University of Edinburgh, Edinburgh, UK. [3]Department of Paediatric Respiratory and Sleep Medicine, Royal Hospital for Children and Young People, Edinburgh, UK. [4]Hospices Civils de Lyon, Hôpital de la Croix-Rousse, Centre de Biologie Nord, Institut des Agents Infectieux, Laboratoire de Virologie, Lyon, France. [5]Department of Pediatrics, Section of Infectious Diseases, University of Colorado School of Medicine and Children's Hospital Colorado, Aurora, CO, USA. [6]West of Scotland Specialist Virology Centre, NHS Greater Glasgow and Clyde, Glasgow, UK. [7]Department of Ecology and Evolution, University of Chicago, Chicago, IL, USA. [8]Fogarty International Center, National Institutes of Health, Bethesda, MD, USA. ✉e-mail: ehowerton@princeton.edu

because they are subject to different seasonal forcing (i.e., seasonal changes in transmission rates due to variation in factors such as climate or human behavior[10,11]). Alternatively, because RSV and hMPV both infect children, viral interactions between these pathogens could drive differences in outbreak timing.

There are a range of possible ways that these interactions could occur, including through ecological mechanisms related to behavior change of infected hosts or through within-host mechanisms that alter host susceptibility, viral load, infectious period, or disease severity[12,13]. Although both negative and positive mechanisms are theoretically possible[14], RSV-hMPV interactions are generally hypothesized to be negative (i.e., hampering transmission or disease severity) and asymmetric (i.e., RSV affects hMPV more strongly than vice versa). Prior studies have investigated different possible within host mechanisms. For example, when single and dual infections were incubated in tissue culture, hMPV replicated more slowly than RSV and was more susceptible to interferon gamma responses[15], pointing to the plausibility of within-host competition for cells and innate immunity-mediated interactions. In this study, RSV replication rate was unchanged by the presence of an hMPV coinfection or infection 2 days prior. Cross-neutralizing antibodies have been discovered, which likely interact via two similar sites on the postfusion F protein[16–19]; however, it is unclear whether these antibodies provide cross-protection across the population. In studies of both human adults[20] and mice[21], antibodies elicited by RSV or hMPV infection did not protect against infection with the other. Nevertheless, one population-level study of more than a decade of respiratory virus hospitalizations in Utah, USA found evidence for a cross-protective effect between RSV and hMPV, where immunity from RSV infection was estimated to substantially reduce the rate of subsequent hMPV infection, but hMPV cross-protection against subsequent RSV infection was estimated to be weak[12].

In systems with interactions between pathogens, pharmaceutical interventions targeted at one pathogen or serotype can have unintended consequences for off-target pathogen dynamics[22]. For example, pneumococcal vaccination against a subset of serotypes enabled an increase in prevalence of other serotypes; a similar effect has been postulated in China for enteroviruses that cause hand, foot and mouth disease[23,24]. This phenomenon of changing dominant serotypes was coined strain replacement[25] and has parallels with competitive release in ecology[26], where reducing the density of one species increases the growth rate of another species that is competing for the same resource.

Given the possibility of interactions between RSV and hMPV, we speculate that the ongoing rollout of new RSV interventions[27–30] could have unintended consequences for hMPV dynamics, and here we quantify the likelihood of this outcome. We present a two-strain mathematical model to capture the effect of potential interactions between RSV and hMPV, and we use historical time series data on detections of RSV and hMPV in Scotland to estimate interaction strength. Importantly, we distinguish the effects of pharmaceutical interventions that target a particular pathogen or strain with those of non-pharmaceutical interventions (NPIs, e.g., social distancing or mask wearing in response to the COVID-19 pandemic), which typically have a non-specific effect on transmission. Leveraging this distinction, we test the plausibility of our model by its ability to explain post-pandemic rebound dynamics of both pathogens simultaneously, and we explore whether our model can also reproduce qualitative patterns observed in Korea and Canada. The disruption of RSV and hMPV transmission by NPIs implemented during the COVID-19 pandemic[31,32] provides a unique opportunity for out-of-sample validation of our interaction model. Finally, we use our model to anticipate potential changes in hMPV dynamics upon rollout of RSV interventions.

## Results

### Variation in RSV and hMPV outbreak patterns across multiple locations

We collected historical time series of weekly RSV and hMPV detections from Scotland, Korea, and Canada (Fig. 1). Across these locations, RSV and hMPV exhibit annual outbreaks where hMPV outbreaks consistently lag RSV outbreaks (Supplementary Fig. 2). Pre-pandemic RSV and hMPV detections are most highly correlated at a lag of 5 weeks (correlation = 0.51, 95% confidence interval, 0.45–0.56) in Scotland. Lags in Korea are substantially longer (19 weeks, correlation = 0.78, 0.72–0.82). In Canada, three regions also exhibit annual epidemics (lags ranging from 3 to 7 weeks); however, in British Columbia and the Prairies region (which includes Manitoba and Saskatchewan), RSV and hMPV seemingly exhibit major/minor biennial outbreaks, although the available historical data is not long enough for such signals to be verified through time series analysis (Supplementary Fig. 2). In regions that appear to exhibit biennial patterns before the pandemic, the timing and magnitude of hMPV outbreaks are negatively associated with the size of the corresponding RSV outbreak. In other words, hMPV outbreaks are comparatively smaller and later in the season in years with a large RSV outbreak, whereas hMPV outbreaks are larger and

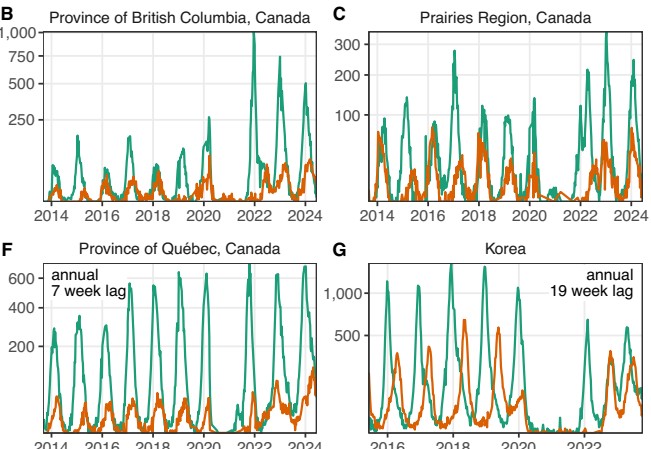

**Fig. 1 | Historical time series of RSV and hMPV.** Detections of RSV (green) and hMPV (orange) detections in (**A**) Scotland, (**B**–**F**) five public health regions of Canada, and (**G**) Korea. Y-axis is on square root scale, and the magnitude varies for each location. For annual outbreaks, the lag reported is that which maximizes cross-correlation between the RSV and hMPV pre-pandemic detections. Although British Columbia and Prairies Region in Canada appear to exhibit major/minor biennial outbreaks by visual inspection, available historical data before the pandemic does not go far enough back in time to verify this statistically. See Supplementary Fig. 1 for results from full cross-correlation analysis and Supplementary Fig. 2 for cross-wavelet transform analysis.

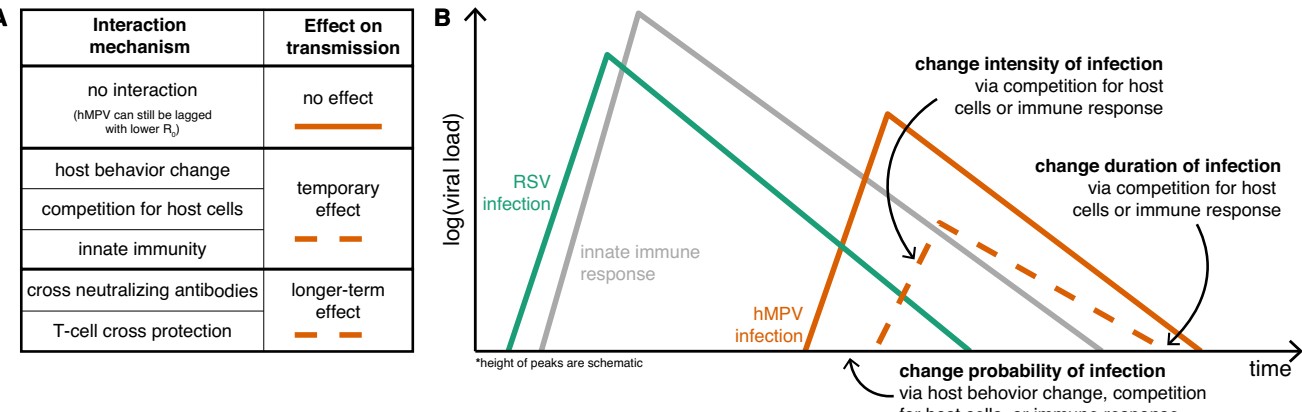

**Fig. 2 | Conceptual schematic of possible RSV and hMPV interaction mechanisms. A** Six possible interaction mechanisms and the associated duration of the interaction effect. For example, innate immunity could confer protection against secondary infection up to a few days after the primary infection (temporary effect), whereas adaptive immunity (antibody or t-cell cross protection) could confer protection against secondary infection weeks or months after the primary infection (longer-term effect). **B** Schematic within host dynamics of RSV infection (green), the innate immune response that is stimulated (gray), and a potential subsequent hMPV infection (orange). An hMPV infection without an interaction effect (solid orange line) may still be more likely to occur after RSV infection if hMPV is less

transmissible than RSV. Interactions between pathogens could also change the probability, intensity, or duration of hMPV infection (dashed orange line). Interaction mechanisms with a temporary and longer-term effect are both shown with a dashed orange line because we hypothesize that the same effects on transmission are possible in each scenario, but they differ in how long after primary RSV infection the effects could manifest (i.e., an hMPV infection far enough along the time axis may be protected only by longer-term mechanisms and not by temporary mechanisms). Although negative effects on hMPV infection are illustrated (e.g., reducing intensity or duration of infection), positive effects could also be possible.

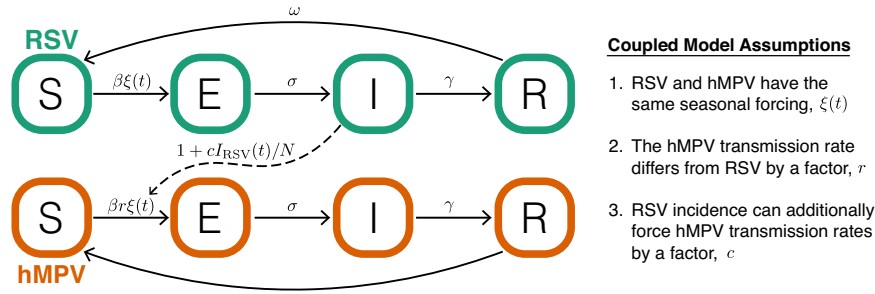

**Fig. 3 | Compartmental diagram of coupled SEIRS (susceptible-exposed-infected-recovered-susceptible) model.** Individuals in the population are categorized based on their infection status to RSV (green) and hMPV (orange). New infections occur based on a baseline transmission rate, $\beta$, that varies seasonally $\xi(t) = (a \cos(2\pi(w(t)/52 - p)) + 1)$, and can differ for hMPV based on some factor $r$. In the coupled model, the seasonal forcing function $\xi(t)$ is assumed to be the same for RSV and hMPV, and RSV incidence, and $(1 + cI_{RSV}(t)/N)$ can additionally force the

hMPV transmission rate. After infection, individuals temporarily remain in the exposed class, where they are not yet infectious. Exposed individuals (E) become infectious (I) at rate $\sigma$, and infectious individuals recover (R) at rate $\gamma$. Recovered individuals lose immunity and become susceptible (S) at rate $\omega$. Versions where summed RSV incidence over some lag can force hMPV transmission were also considered (i.e., $\sum_{i=0}^{l} I_{RSV}(t - l)$) as in Supplementary Information Section 2.2).

earlier in years with a small RSV outbreak (as suggested by changing phase across time in cross wavelet transform analysis, Supplementary Fig. 2). Across locations, the timing and magnitude of post-pandemic resurgences vary substantially.

### Population-level model to characterize the presence of potential RSV–hMPV interactions

A range of mechanisms could explain these observed lags (Fig. 2), and we developed an SEIRS transmission model that includes multiple possible mechanisms (Fig. 3). This model is characterized by three key assumptions. First, later hMPV outbreaks could be driven by mechanisms unrelated to RSV (no interaction hypothesis, Fig. 2). For example, a lower transmission rate for hMPV means the outbreak grows more slowly and will peak later. Thus, we let the hMPV baseline transmission rate differ from RSV by some factor, $r$. Second, the phase of seasonal forcing is often used to capture outbreak seasonality. Given similarities in pathogen biology, we assume that RSV and hMPV

are subject to the same seasonal forcing, including when seasonal forcing peaks (timing) and how much stronger transmission is at the peak (amplitude). We also consider a null model where seasonality can differ for each pathogen.

Finally, we include the possibility for prior RSV infection to affect hMPV dynamics. Conceptually, a range of mechanisms is possible, and we propose to group these mechanisms based on the duration of the interaction effect (temporary and longer-term interaction hypotheses, Fig. 2). For example, behavioral change, competition for host cells during coinfection, and innate immunity could all confer protection against secondary heterologous infection on a short timescale. This contrasts with adaptive immunity, including antibody or T-cell cross-protection, which could affect secondary infection for a longer period after primary infection. We hypothesize that these interaction mechanisms could alter the likelihood, intensity, or duration of secondary infection, although each mechanism may not alter each component equally. Changes in the probability of reporting could also be possible.

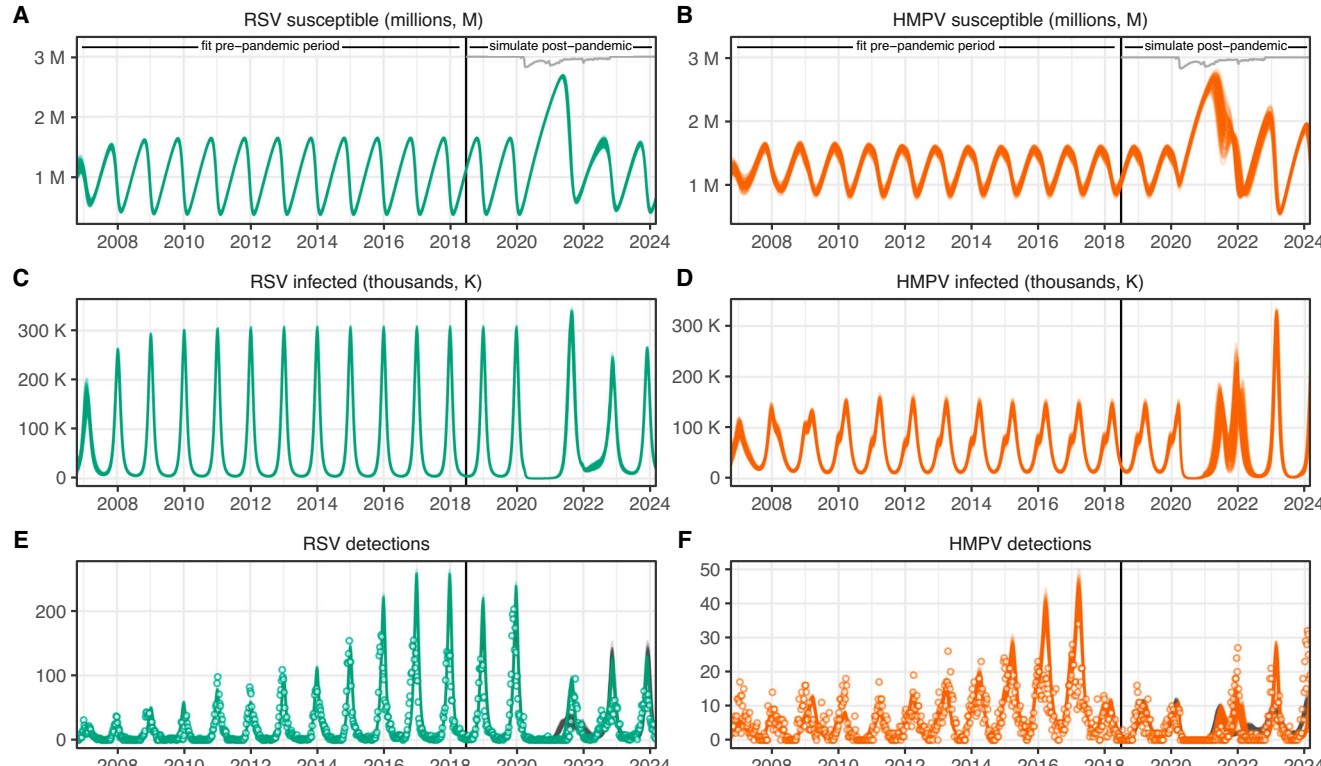

**Fig. 4 | Fitted model results for RSV (green) and hMPV (orange) in Scotland from week of October 16, 2006, through week of March 4, 2024.** The model estimates the number of susceptible individuals (**A**, **B**), the number of infected individuals (**C**, **D**), and the number of detected infections (**E**, **F**). Model estimates of detections account for underreporting and variations in testing (see Supplementary Fig. 3). The model was fit to observed detections of RSV and hMPV (points in **E**, **F**) before week of June 16, 2018 (black vertical line) and simulated forward out of sample, using Google mobility data (gray) to determine the strength of NPIs during the pandemic perturbation. One hundred random samples from the posterior distribution are shown; one hundred random out of sample post-pandemic simulations from the independent model are also shown for reference (black). See Supplementary Fig. 5 and Supplementary Fig. 6, respectively, for results from a sensitivity analysis using an SIRS model and independent SEIRS model with no interaction between RSV and hMPV.

Interactions that cause changes in susceptibility or transmissibility can be modeled as a scalar effect on the transmission term[33]. Given that we do not have strong evidence for a particular interaction mechanism, we modeled potential interactions through this simple scalar on transmission (Methods, Fig. 3). In particular, we assume that RSV incidence can additionally force hMPV transmission rates by some factor, $c$. We take $c$ not significantly different from zero to imply no evidence for an interaction (i.e., no effect in Fig. 2), and $c < 0$ to imply RSV infection reduces hMPV transmission rate (i.e., RSV infection makes individuals less susceptible to hMPV infection or makes future hMPV infection less transmissible). Our assumption of asymmetric interactions is based on evidence from prior modeling[12] and experimental studies[15].

To test which mechanisms are consistent with observed outbreak dynamics, we fit this model to RSV and hMPV detection data in Scotland, given the long historical timeseries available. White et al.[34] explained variation in dynamical patterns of RSV A/B using a two-strain cross-immunity model; however, because data on RSV A/B were unavailable, our model captures interactions between RSV (totaled across subtypes) and hMPV. We fit the model using pre-pandemic outbreaks (from October 2006 through June 2018), where Bayesian parameter estimation was performed via the Hamilton Monte Carlo algorithm. To test the fitted model out of sample, we simulated outbreaks from July 2018 through March 2024. This period was chosen to include one season before as well as the period during and after the COVID-19 pandemic; we assumed that COVID-19 NPIs reduced RSV and hMPV transmission rates proportional to observed reductions in mobility[35]. We began by using current RSV incidence to characterize the presence

of an interaction, and then consider lagged versions of RSV incidence to approximate different durations of an effect. See Methods for additional details on model structure, fitting, and simulation.

The fitted SEIRS model captured the pre-pandemic seasonal outbreak patterns for RSV and hMPV; it was also able to reproduce out of sample the general timing and magnitude of post-pandemic rebounds (Fig. 4). Parameter estimates suggest hMPV has a 17% lower transmission rate compared to RSV ($r = 0.83$, 90% credible interval, CI: 0.8–0.87), and a suppressive effect of RSV incidence on hMPV transmission rate ($c = -7.13$, 90% CI: −7.76 to − 6.49). Much of the year-to-year variation in detections of RSV and hMPV outbreak magnitude before the pandemic were explained by variations in testing, which we accounted for in the model with data on the number of tests that were performed (Methods, Supplementary Fig. 3). The number of tests performed decreased after the pandemic, yet post-pandemic outbreaks were predicted to be larger due to a buildup of susceptible individuals during the pandemic NPI period[36] (number of susceptible and infected individuals, Fig. 4). In an SIRS model with interaction, similar qualitative results were obtained (Supplementary Fig. 5), although the interaction between RSV and hMPV was estimated to be weaker ($c = -3.82$, 90% CI: −4.25 to −3.38, Supplementary Fig. 4).

We also compared performance to an alternative null model where RSV and hMPV are assumed to be independent (i.e., no interaction between pathogens and relaxing the assumption of identical seasonality, Fig. 5). The independent model estimated peak seasonal forcing to be earlier for RSV and later for hMPV than the coupled model, as expected given the lags between outbreaks (Supplementary Fig. 4). By adjusting seasonality, the independent model was able to

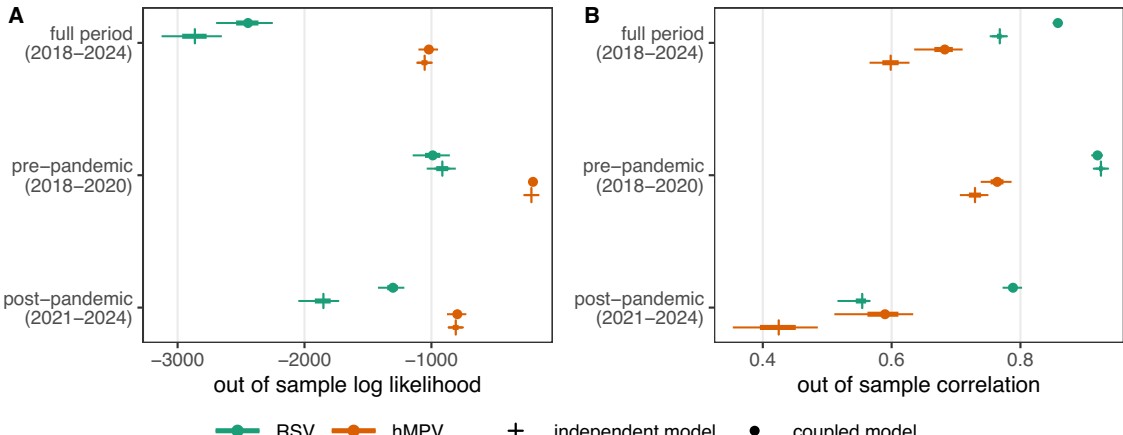

**Fig. 5 | Out of sample performance for coupled and independent model fits.** For both models, each sample from the posterior distribution is used to simulate the model forward from June 25, 2018 until March 4, 2024. Simulations are compared to observations using (**A**) log likelihood and (**B**) correlation on a log scale; the distribution of performance across $n = 7500$ posterior samples is summarized by Q5-Q95 (thin line), Q25-Q75 (thick line), and Q50 (circle or cross). The out-of-sample period was chosen to include one full pre-pandemic season (beginning in 2018), as well as all seasons during and after the COVID-19 pandemic perturbation. Performance is summarized across the full out of sample period, as well as for the pre- and post pandemic portions of the out of sample period, and for both pathogens (RSV in green, hMPV in orange). The pre-pandemic portion of the out of sample period is defined from June 25, 2018–March 1, 2020, and the post-pandemic portion of the out of sample period is defined from January 1, 2021–March 4, 2024; weeks in 2020 after March 1 were excluded to avoid artificially inflating performance with zeros. The coupled model (circle) assumes that RSV and hMPV are subject to the same seasonality, and that RSV infection can affect the transmission of hMPV. The independent model (cross) relaxes both of these assumptions, with independent seasonalities and no interaction. See Supplementary Fig. 6 for full results from the independent model.

reproduce pre-pandemic RSV and hMPV dynamics out of sample as well as the coupled model. However, performance of the independent model was worse when predicting post-pandemic rebounds. For example, during pre-pandemic out of sample seasons (June 24, 2018–March 1, 2020), the correlation between observations and simulations for RSV was 0.920 (90% CI: 0.910–0.928) from the coupled model versus 0.925 (90% CI: 0.917–0.932) from the independent model; for hMPV, correlation was 0.764 (90% CI: 0.738–0.787) and 0.730 (90% CI: 0.707–0.751). In contrast, for all weeks after January 1, 2021, the median out of sample correlation for RSV was 0.789 (90% CI: 0.772–0.802) from the coupled model and 0.554 (90% CI: 0.517–0.567) from the independent model, and 0.591 (90% CI: 0.510–0.634) versus 0.424 (90% CI: 0.352–0.486) for hMPV. When using log likelihood to assess performance, there was no significant difference between post-pandemic predictions for hMPV. Differences between the coupled and independent models vary through the out-of-sample period and depend on assumptions about NPIs (Supplementary Fig. 16).

In this model, the term $c$ represents the effect of a single RSV infection on the transmission rate of hMPV. One way to interpret this parameter is the reduction in the pool of hMPV susceptible population for each new RSV infection. For example, in a population of individuals completely susceptible to hMPV, we estimated that a single RSV infection temporarily prevents seven new hMPV infections (i.e., $c = -7.13$). Given heterogeneous mixing, for example, a single RSV infection could hypothetically reduce the pool of individuals susceptible to hMPV by more than one individual; however, per-capita effects much larger than one are biologically unlikely. This large per-capita effect likely arose because our model assumed that only current RSV incidence forces hMPV transmission. To test this, we re-estimated parameters using longer lags in RSV incidence to force hMPV transmission (e.g., sum of the prior 6 weeks of RSV incidence, Supplementary Information Section 2). As lags increase, the per-capita effect gradually approaches 1, with an estimated per-capita effect of $-1.03$ (90% CI: $-1.12$ to $-0.94$) for a 7-week lag. Longer lags were not considered, as there is a higher chance of double-counting RSV reinfections. Estimates of other biological parameters did not change significantly (Supplementary Figs. 11, 12) and model performance

(Supplementary Fig. 14) remained consistent, suggesting minimal additional difference between these versions of the model. As such, we use the simple model without lagged incidence for all subsequent analyses.

## Implications for hMPV burden and dynamics under RSV interventions

Although limited historical data prevented us from also fitting our model to Korea or regions in Canada, we explored whether the fitted Scotland model could produce the qualitative patterns observed in these locations, including longer lags between outbreaks of the two infections in Korea and early/late biennial patterns in regions of Canada (Fig. 6). To do so, we created a bifurcation diagram of asymptotic dynamics as a function of the seasonal forcing parameter, given that variation in seasonal transmission is known to drive dynamical differences between locations for RSV[9] and other pathogens[37]. Because we do not have strong hypotheses about the seasonality expected in these locations, we tested a wide range of seasonal forcing and explored the resulting outbreak dynamics. For weak seasonal forcing (amplitude, $a < 0.6$), RSV and hMPV both exhibit annual outbreaks, and the lag between outbreaks increases as seasonal forcing becomes weaker. For example, low levels of seasonal forcing exhibit a longer lags between outbreaks, similar to the dynamical patterns observed in Korea. For higher seasonal forcing, outbreaks can exhibit biennial, 3-year, or chaotic cycles. In biennial regimes, the timing of the outbreak peaks alternates between early and late in the season.

For each of these dynamical regimes, we also simulated potential effects of the introduction of RSV interventions. Since the effects of RSV interventions on the RSV-hMPV interaction are uncertain, we chose to model a generic worst case scenario from the perspective of hMPV public health burden, where immunity induced by a given RSV intervention is assumed to have no protective effect against future hMPV infection, and RSV intervention coverage is complete. In other words, the suppressive effect of RSV infection on the hMPV transmission rate has been completely eliminated (i.e., $c = 0$ versus the fitted value $c = -7.13$; dashed orange lines, Fig. 6C). For example, under the worst case scenario in Scotland, the model predicted hMPV outbreaks

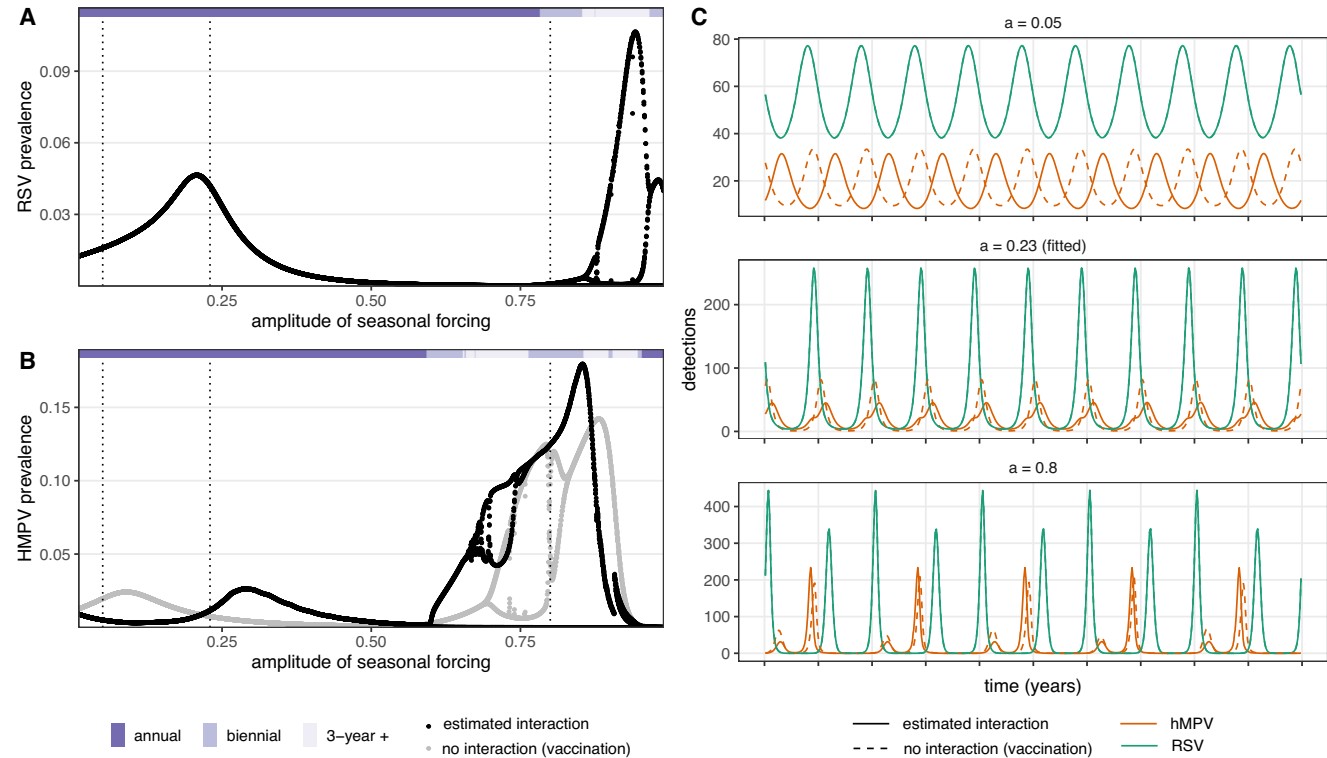

**Fig. 6 | Affect of seasonal forcing on coupled dynamics.** Bifurcation diagrams for RSV (**A**) and hMPV (**B**) asymptotic dynamics in the fitted model. The prevalence of each pathogen on the 12th week of the year (*y*-axis) is shown as the amplitude of seasonal forcing increases. For hMPV, a second bifurcation diagram is shown for the worst-case vaccination scenario (gray dots), where RSV infection does not protect against subsequent hMPV infection at all (*c* = 0). Seasonal regimes of the fitted model are shown with purple bars at the top of each bifurcation diagram (annual: dark purple, biennual: medium purple, 3-year cycle and chaotic regimes: light purple). Dotted vertical lines correspond to seasonal forcing values in three example time series shown in (**C**). In each example, equilibrium dynamics for RSV (green) and hMPV (orange) are shown, where hMPV dynamics in the fitted model (solid lines) are contrasted to the worst-case vaccination scenario (dashed lines).

would shift to be 5 weeks earlier (90% CI: 4–6 weeks) with an 80% larger peak (36 more detections at the peak, 90% CI: 32–42), although the annual burden of hMPV would not change substantially (1.6% increase, 90% CI: 1.4–1.8%). There is estimated to be a 20% probability that hMPV peaks double; yet, the model predicts that hMPV outbreak peaks will not exceed pre-intervention RSV peaks (Supplementary Fig. 8). To approximate changes in mean age of infection under the worst case scenario, we also incorporated heterogeneous, age-structured mixing into our fitted model (Supplementary Information Section 2). This suggested that eliminating the suppressive effect of RSV infection on hMPV transmission would decrease the mean age of hMPV infection (e.g., by ~3 weeks in children under 3 years of age, Supplementary Fig. 10). Predicted changes in outbreak size, peak timing, and peak magnitude depended non-linearly on the RSV-hMPV interaction term, *c*, and on the strength of seasonal forcing; for example, in locations with seasonal forcing lower than Scotland, the timing of hMPV outbreaks could shift more dramatically. However, even in the scenario with the largest increases, hMPV burden and peak magnitude were again not predicted to exceed pre-intervention RSV outbreaks (Supplementary Fig. 9).

## Discussion

Here, we investigated potential interactions between two important respiratory pathogens, RSV and hMPV, by analyzing patterns in their population-level outbreak dynamics. The epidemiological time series from Scotland, Canada, and Korea demonstrate qualitatively different coupled dynamics, including lagged annual outbreaks and out-of-phase biennial outbreaks. Our simple model can reproduce these patterns, where the best-fitting parameters suggest lower transmissibility of hMPV and a suppressive effect of RSV infections on hMPV

transmission. In a worst-case scenario where new RSV interventions eliminate suppressive effects on hMPV transmission, the model predicts that increases in the peak size of hMPV outbreaks and changes in timing are possible. Yet, these changes are not predicted to exceed pre-intervention RSV burden, and thus, RSV interventions are expected to decrease overall combined burden.

Our phenomenological model, which assumed hMPV transmission dynamics were coupled to RSV incidence, better captured post-pandemic rebound dynamics than the null model that assumed independent RSV and hMPV outbreak dynamics. The pandemic perturbation provided an additional test to help distinguish between these two hypotheses, as pre-pandemic patterns could be captured in the independent model with different seasonal forcing of the transmission rate for each pathogen. Other studies have also used population-level epidemiological data to make inferences about viral interactions, with mixed findings for RSV and hMPV. Correlational analyses of monthly viral incidence have found no evidence of interactions[38,39] or evidence of weak positive interactions, where prior infection facilitates secondary infection[40]. We hypothesize that these findings contrast with ours (Nickbakhsh et al.[40] even used a subset of the same Scottish data) because the correlational analyses used monthly incidence data, which could mask the short-term interactions we have modeled, although further investigation is warranted. Analyses of coinfection patterns have also provided contradictory results[40,41]; however, inferring interactions from coinfection prevalence has been shown to be difficult to interpret and biased in many settings[42,43]. Nonetheless, our results reinforce the idea that mechanistic models offer promise in explaining polymicrobial outbreak patterns[12,44].

Given inferred interactions between RSV and hMPV, we also used our model to quantify the likelihood of hMPV surges in the face of RSV

interventions. In addition to the ongoing use of monoclonal antibodies and maternal vaccines which have demonstrated efficacy in protecting infants[28–30,45], there are compelling reasons to develop a pediatric RSV vaccine that extend beyond direct protection of children against RSV disease[46,47]. There are strong links between RSV infection and pneumococcal respiratory tract infections[48–50], suggesting that reduction in RSV burden may offer additional benefits against secondary bacterial infection for children. Moreover, if pediatric vaccines slow RSV transmission, they could offer indirect protection for seniors[51] and other high-risk groups. However, increases in hMPV burden could counteract some of the benefits of RSV interventions and reduce cost-effectiveness. Continued development of RSV pharmaceutical interventions, including RSV-hMPV vaccines, offers promise in offsetting this challenge[52]. In addition, understanding the likelihood of increases in hMPV will be important for the robustness of test-negative vaccine efficacy designs[53].

To assess the possibility of hMPV surges, we made a crude worst-case assumption from the perspective of hMPV burden, where RSV interventions completely eliminate the suppressive effects of viral interactions on hMPV transmission. The likelihood of such an effect depends on how RSV interventions affect possible RSV-hMPV interaction mechanisms. For example, if interactions occur primarily through host behavior change, interventions that prevent RSV infection or reduce severe disease would eliminate the interaction effect (as host behavior change is most likely driven by disease severity). However, given high rates of mild or asymptomatic infection[54], we hypothesize that behavioral change is unlikely to be the sole driver of interactions. Within host interactions, including competition for host cells, or host immunity (innate or adaptive) seem more plausible. Compared to current interventions, pediatric vaccines may be more likely to prevent RSV infection and transmission[55,56]; however, it is not clear whether passive immunization (through monoclonal antibodies, maternal vaccines) or active immunization (through pediatric vaccination) could engender the most substantial change in RSV-hMPV interaction mechanisms. Moreover, our assumption implies perfect coverage of RSV interventions, yet realized coverage will be imperfect. For these reasons, we believe the effects of all RSV interventions will fall somewhere between the two pathogen interaction scenarios we have modeled.

Even under our conservative, worst-case assumption, overall hMPV burden and peak incidence is predicted to be lower than pre-intervention RSV burden. This result could be driven by at least two factors. First, our model estimated that hMPV was less transmissible than RSV, consistent with a higher mean age of infection for hMPV[57–59]. This inherent difference in transmissibility limits the magnitude of hMPV outbreaks, even in the absence of RSV cross-protection. In addition, the model predicted lower reporting of hMPV infections compared to RSV, although the estimate of hMPV reporting rate may be unrealistically low. More significant under reporting may be due in part to lower severity of hMPV compared to RSV, especially in the youngest children[57]. However, if RSV interventions shift hMPV burden to younger infants, the number of severe cases could increase. RSV-hMPV dynamics in coming seasons will enable us to test our predictions beyond early evidence that is available[60]. The global policy implications of such predictions should also be explored.

Our model is subject to a number of limitations. First, our quantitative estimates of interactions between RSV and hMPV should be interpreted with caution. Important factors that could affect the interaction between RSV and hMPV may be omitted in our simple model. We used a single scalar term, $c$, to phenomenologically capture all interactions between RSV and hMPV. While this approach provides useful foundation for detecting possible interactions, it risks over-simplifying complex biology, such as immunological mechanisms of protection and the changing intensity of protection over time. In addition, our framework cannot easily detect the duration of viral interactions. We showed that lagged 7 weeks of past RSV incidence yielded a 1-1 per-capita effect of a single RSV infection in decreasing the transmission rate of hMPV. Per-capita effects of less than one from longer-duration interactions are plausible but difficult to interpret in this framework, given how we model reinfections. Future modeling work should focus on quantifying the duration of interaction as a means to differentiate innate and adaptive mechanisms. Although we accounted for variation in testing across time and between the two pathogens, changes in probability of care-seeking with RSV-hMPV coinfection, for example, could also affect the observed dynamics. Moreover, although the independent model did not reproduce post-pandemic rebounds as well as the coupled model, other goodness-of-fit measures may need to be considered, and this hypothesis should be tested in other locations.

Second, we modeled transmission dynamics using an SEIRS model with homogeneous mixing, although more complicated structures have been proposed for RSV[8,56]. We used a long estimate for immune waning from a prior study[34], which could conflict with observations of frequent reinfections[54]; however, long waning is likely consistent with our data due to the nature of syndromic surveillance data, where reporting of asymptomatic or mildly symptomatic reinfections is rare. We introduced an age-structured extension of our model; however, we fit only to aggregated data without age structure. Transmission and severity of RSV and hMPV are strongly age dependent[61,62], and thus age-specific patterns could affect our conclusions in important ways (including our estimate of change in mean age of hMPV infection without RSV interactions). Furthermore, our worst-case scenario assumes that RSV interventions would affect the entire population, yet RSV interventions will almost certainly target specific age groups[52]. Future work should explore the implications of these complex age-specific patterns on RSV-hMPV interactions in depth.

Finally, issues with parameter identifiability are a potential concern for our model. Given known identifiability issues with epidemiological models, including the SEIR model[63,64], we fixed biological parameters available in the literature. Although this could affect our conclusions, we believe the post-pandemic rebounds provide an out-of-sample test for our model fits. Nevertheless, this test depends on assumptions about NPIs, which are highly uncertain, and it is still possible that alternative parameter sets could fit pre- and post-pandemic dynamics equally well. We could not model the dynamics of RSV A and B separately, which are known to have complex dynamics that could affect model predictions[34], and we did not consider potential interactions with other pathogens such as influenza. Repeating this analysis in other geographic locations will be one important test of robustness. Although we were only able to fit our model in Scotland due to short historical time series in other locations, fitting the model using pre- and post-pandemic data (e.g., as proposed in ref. [65]) could overcome this challenge and take advantage of more recent efforts to sequence and report RSV infections by subtype.

Future work should continue to explore variation in RSV-hMPV outbreak patterns across locations, as such differences may provide insight into the mechanisms of RSV-hMPV interactions. For example, the consistency of annual, lagged outbreaks of RSV and hMPV stands in contrast to the multi-annual cycles observed for RSV subtypes[34,66]. The early/late outbreak timing observed in biennial regions of Canada has also been reported for hMPV in the United States[67] and central Europe[41]; although these reports do not directly relate the pattern in hMPV outbreak timing to RSV, such patterns seemingly provide evidence for viral interactions, but more investigation is needed. Finally, the longer lags exhibited in Korea need to be better understood. Although it is possible these lags are explained by weaker seasonal forcing, other mechanisms such as low birth rates and differences in reporting should also be considered. As we have used here, a crucial test for different mechanistic hypotheses is whether they can

reproduce rebound outbreak dynamics after the COVID-19 pandemic, especially across a range of locations[68,69]. Coupling incidence data with coinfection data[40], longitudinal serological data[70–72], and antigenic maps[73] (that include RSV A, RSV B, and hMPV) will ultimately provide the strongest evidence for mechanisms that underlie pathogen interactions and enable coexistence. Moreover, our results highlight the possibility that multi-pathogen models could offer improved predictive power. Understanding and anticipating the cases in which polymicrobial interactions drive dynamics will be essential for future outbreak prediction efforts.

Here, we have leveraged a mechanistic transmission model to explain variation in coupled RSV-hMPV outbreak patterns, and we used post-pandemic rebound outbreaks as an out-of-sample test for our model. We found evidence for interactions between RSV and hMPV, where RSV infection temporarily reduces the probability of hMPV transmission at the population level. Our model predicted that changes in peak timing and magnitude of hMPV outbreaks are possible in the face of RSV interventions; however, hMPV burden is not predicted to exceed pre-intervention RSV burden, likely because hMPV was estimated to be less transmissible than RSV. Continuing to explore these polymicrobial outbreak patterns will shed light on mechanisms of viral interaction, with implications for prediction of future outbreak dynamics and development of new therapeutics.

## Methods

### Data

We collected historical time series of weekly RSV and hMPV detections in Scotland, Canada, and Korea. In Scotland, samples from patients with respiratory illness were collected in primary and secondary care settings and submitted to the West of Scotland Specialist Testing Center for PCR testing. Samples were primarily collected from the west of Scotland and the Glasgow area, although a small subset of national surveillance samples from across Scotland were collected annually. Until 2021, these surveillance samples composed a small portion of all tests performed, but large increases were seen thereafter due to new surveillance protocols. We aggregated data on the number of tests performed and the number of positive tests for each pathogen into a weekly time series from October 9, 2006 to March 4, 2024. Birth rate and population size were approximated from the National Records of Scotland demographic data (https://www.nrscotland.gov.uk/statistics-and-data/births-deaths-marriages-and-life-expectancy/#).

We also scraped data on respiratory virus testing and detections from the Public Health Agency of Canada respiratory virus weekly reports (https://www.canada.ca/en/public-health/services/surveillance/respiratory-virus-detections-canada.html), which are publicly available online starting in the 2013–2014 season. Data are published for each reporting laboratory in Canada and aggregated across the Canadian public health regions. We identified an issue with RSV reporting in the Province of Alberta, so we excluded this province from the Prairies region. Finally, we collected data on RSV and hMPV detections in Korea from publicly reported sentinel surveillance data provided by the Korea Disease Control and Prevention Agency website (https://dportal.kdca.go.kr/pot/index.do). In addition, we used Google community mobility reports as a proxy for relative intensity of transmission control due to COVID-19 non-pharmaceutical interventions (https://www.google.com/covid19/mobility).

### Transmission model

We model RSV and hMPV transmission dynamics using a simple SEIRS dynamical model, which tracks the infection status of all individuals in the population. Susceptible individuals (S) become exposed (E) based on some transmission rate $\beta$, where they are infected but not yet infectious. Exposed individuals eventually become infectious (I) at some rate $\sigma$, and infected individuals recover (R) at some rate $\gamma$. Recovered individuals are protected from reinfection until their immunity wanes, at rate $\omega$. The model can be written as a system of differential equations for $i \in \{RSV, hMPV\}$:

$$\frac{dS_i}{dt} = \mu(N - S_i(t)) - \lambda_i S_i(t) + \omega R_i(t) \tag{1}$$

$$\frac{dE_i}{dt} = \lambda_i S_i(t) - \sigma E_i(t) - \mu E_i(t) \tag{2}$$

$$\frac{dI_i}{dt} = \sigma E_i(t) - \gamma I_i(t) - \mu I_i(t) \tag{3}$$

$$\frac{dR_i}{dt} = \gamma I_i(t) - \omega R_i(t) - \mu R_i(t) \tag{4}$$

where $\mu$ is the birth and death rate, $N$ is the total population size, and $\lambda_i$ is the force of infection for pathogen $i$. To model the potential interaction between RSV and hMPV, we make the following assumptions:

1. RSV and hMPV transmission rates are subject to the same seasonal forcing $(a\cos(2\pi(w(t)/52 - p)) + 1)$, for week $w(t) \in 1, …, 52$,
2. hMPV transmission rate differs from RSV transmission rate by a factor $r$, and
3. hMPV transmission rate can be forced by observed RSV incidence, $(1 + cI_{RSV}(t)/N)$.

We consider only the effect of RSV on hMPV given the results in Bhattacharyya et al.[12] and Geiser et al.[15] suggesting the effect of RSV on hMPV is stronger than vice versa. Together, these three assumptions yield the following force of infection terms for RSV and hMPV:

$$\lambda_{RSV} = \beta(a\cos(2\pi(w(t)/52 - p)) + 1)\frac{I_{RSV}(t)}{N} \tag{5}$$

$$\lambda_{hMPV} = \beta r(1 + cI_{RSV}(t)/N)(a\cos(2\pi(w(t)/52 - p)) + 1)\frac{I_{hMPV}(t)}{N} \tag{6}$$

We contrast this model to a model where RSV and hMPV are independent and subject to different seasonal forces. In this model, we define the force of infection as

$$\lambda_{RSV} = \beta_r(a_r\cos(2\pi(w(t)/52 - p_r)) + 1)\frac{I_{RSV}(t)}{N} \tag{7}$$

$$\lambda_{hMPV} = \beta_m(a_m\cos(2\pi(w(t)/52 - p_m)) + 1)\frac{I_{hMPV}(t)}{N} \tag{8}$$

In both cases, we use a sine wave to capture seasonal changes in the transmission rate, which is commonly used for respiratory viruses in temperate regions such as Scotland. In addition, we use a single seasonal forcing function for the entire population as the majority of the data comes from Glasgow, decreasing the likelihood that there are major variations in seasonality underlying the observed outbreaks. We also fit a coupled SIRS model to test the sensitivity of our results to model structure; because RSV-hMPV interactions were estimated to be stronger in the SEIRS model, we use this as the primary model in order to create a worst case scenario. Results from the independent SEIRS models and coupled SIRS model are provided in the supplement.

We use a discretized version of the above model for computational efficiency in parameter estimation. In particular, we discretized the model following He et al.[74], which assumes the number of events that occur follows a Poisson process. For a Poisson process occurring at rate $\lambda$, the probability of no events occurring is $e^{-\lambda\Delta t}$, and the probability of at least one event is $(1 - e^{-\lambda\Delta t})$. Thus, the number of individuals that leave compartment $X$ in time step $\Delta t$ is $\Delta X[t] = (1 - e^{-r\Delta t})X[t - \Delta t]$

assuming $r$ is the sum of rates out of $X$ in the ordinary differential equation model. This discretization scheme ensures that the number of individuals in each compartment will always be positive. See Supplementary Information for additional details and the full discretized model.

## Parameter estimation

We fit model parameters in a Bayesian framework using the Hamilton Monte Carlo algorithm via `rstan`[75,76], a package in the R Statistical Software[77]. We assumed that observed detections are subject to some pathogen-specific degree of under reporting, $\rho_i$, and scale linearly with changes in testing. To represent trends in testing, we calculated a 1-year moving average of the number of tests performed each week, $\tau[t]$, and scaled this value from 0 to 1 (Supplementary Fig. 3; see Supplementary Fig. 15 for additional analysis where incorporation of weekly testing trends yielded qualitatively similar model fits and dynamics). To account for potential overdispersion in the disease transmission process[78], we modeled the observed number of detections for each pathogen as $C_i[t] \sim \text{NegBinom}(\rho_i \tau[t] I_i[t], \phi_i)$ for $i \in \{\text{RSV}, \text{hMPV}\}$ and some shape parameter $\phi_i$ which is estimated based on the dispersion of the data; $C_i[t]$ represents weekly counts of RSV and hMPV detections in Scotland, and we defined the fitting period as all weeks before June 16, 2018 in order to include one pre-pandemic season in our out of sample period.

In addition, for fitting the model, we rescaled the $(1 + cI_{\text{RSV}}[t]/N)$ term to $(1 + c'D_{\text{RSV}}[t])$, where $D_{\text{RSV}}[t] = I_{\text{RSV}}[t]/\max_{t \leq t_{\max}}(I_{\text{RSV}}[t])$ and $t_{\max}$ is the last week of the fitting period. Scaling RSV incidence to $[0, 1]$ enables us to enforce a strict lower bound for $c'$ that guarantees the force of infection will always be positive (i.e., $1 + c'D_{\text{RSV}}[t] > 0$ yields $c' > -1$). As discussed in the Results section, more restrictive lower bounds could be enforced based on biological context. We fixed the mean incubation time, the mean recovery rate, and the mean waning rate[34,79], and we assumed a constant birth and death rate that was calculated from demographic data to ensure a stable population size. A full list of parameter assumptions and priors can be found in Table S1. We fit the alternative models following the same approach. For all models, we used four chains and 15,000 iterations; we assessed convergence based on diagnostic plots and a lack of warnings from `rstan` about divergent transitions, r-hat, effective sample size, and maximum tree depth[75].

## Out of sample simulations, sensitivity analysis, and model extensions

Using the posterior distribution obtained from parameter estimation, we simulated the model forward for two purposes: (a) test a model's ability to capture observed dynamics out of sample, including one pre-pandemic season as well as post-pandemic rebounds, and (b) explore equilibrium dynamics under a range of parameter values.

First, to simulate post-pandemic rebounds, we needed to account for the effect of NPIs on transmission. We assumed the force of infection for both RSV and hMPV was scaled by a time-varying constant $\delta[t]$, which was approximated using Google mobility data (https://www.google.com/covid19/mobility). The Google mobility reports included six mobility categories, including retail and recreation, grocery and pharmacy, parks, transit stations, workplaces, and residential. The reports provide a single value for each week in a given location that represents the percent change to the number of visits to locations in each category compared to pre-pandemic baselines. Following Park et al.[80], we created a single weekly mobility metric, $G[t]$, by averaging the percent change from baseline across retail and recreation, grocery and pharmacy, transit stations, and workplaces categories for each week. This value is shown as a gray line in Fig. 4. As of October 15, 2022, the Google mobility data were no longer reported. After this point, we assumed transmission rates return to baseline ($\delta[t] = 1$). Given that the

relationship between the relative change in visits to locations as captured by the Google mobility metric and transmission reduction from NPIs is unknown, we let the effect of NPIs on transmission be modeled as $\delta[t] = sG[t]$. We tested a range of scalars, $s$, in increments of 0.05 and chose the scalar that yielded the optimal log-likelihood across RSV and hMPV post-pandemic time series (Supplementary Fig. 7). While here we assumed that the reduction in transmission is the same for RSV and hMPV, we performed a sensitivity analysis where we relaxed this assumption (Supplementary Fig. 16).

Then, to explore equilibrium dynamics under a range of parameter values, we simulated the model for 40 years, assuming no pandemic NPIs and no time-varying changes in testing. We used median estimates for each parameter from the posterior distribution, and for a sensitivity analysis, we varied the amplitude of seasonal forcing from 0 to 1. We repeated this sensitivity analysis for a worst-case vaccination scenario, where the effect of RSV infection on hMPV transmission is eliminated (i.e., $c = 0$).

In addition to the sensitivity of our model to a range of parameter values, we also considered two extensions of the model. First, we tested a model with explicit age structure and heterogeneous mixing using the POLYMOD contact study[81]. Using parameters estimated for the simple homogeneous mixing model, we tested how mean age of infection of hMPV changes with and without interaction with RSV. Second, we considered the possibility that hMPV transmission rate is forced not by current RSV incidence, but a lagged sum of RSV incidence (i.e., RSV incidence over the prior $n$ weeks). A full description of these model extensions and the corresponding results is presented in Supplementary Information Section 2).

## Reporting summary

Further information on research design is available in the Nature Portfolio Reporting Summary linked to this article.

## Data availability

The RSV and hMPV outbreak data from Scotland are available upon request to NHSGGC/NHS Scotland (https://www.informationgovernance.scot.nhs.uk/pbpphsc/home/for-applicants/). Data on weekly incidence of RSV and hMPV for Canada and Korea were collected from publicly available sources. Data for Canada were scraped from the historical Respiratory Virus Detection Surveillance System reports provided by Public Health Agency of Canada, available at https://www.canada.ca/en/public-health/services/surveillance/respiratory-virus-detections-canada.html. Data for Korea were downloaded from the Acute Respiratory Infection section of the Korea Disease Control and Prevention Agency Infectious Disease Statistics website, available at https://dportal.kdca.go.kr/pot/index.do. Demographic data for Scotland was downloaded from the Weekly Births in Scotland report provided by National Records of Scotland (https://www.nrscotland.gov.uk/statistics-and-data/births-deaths-marriages-and-life-expectancy/#). Google mobility data for the UK was downloaded from the Community Mobility Reports provided at https://www.google.com/covid19/mobility/. All data, including raw data when available and posteriors derived from model fitting, can be found in the `/data` folder of the GitHub repository available at https://github.com/eahowerton/hmpv-rsv-intervention and archived at https://zenodo.org/records/15778459. Source data for figures are available as `.rda` files or can be generated using scripts within the repository.

## Code availability

All code used to perform this analysis can be found in a GitHub repository available at https://github.com/eahowerton/hmpv-rsv-intervention and archived at https://zenodo.org/records/15778459. For a complete list of packages used and corresponding versions, see the `renv.lock` file in this repository. Synthetic data to test code can be provided upon request to the corresponding author.

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

## Acknowledgements

The authors thank Bjarke Frost Nielsen and Stanca Ciupe for their thoughtful comments on the model. E.H., B.G., and C.J.E.M. have been funded in whole or in part with Federal funds from the National Cancer Institute, National Institutes of Health, under Prime Contract No. 75N91019D00024, Task Order No. 75N91023F00016. The content of this publication does not necessarily reflect the views or policies of the National Institutes of Health or the Department of Health and Human Services, nor does mention of trade names, commercial products or organizations imply endorsement by the U.S. Government. S.W.P. acknowledges support from the Charlotte Elizabeth Procter Fellowship of Princeton University, and Peter and Carmen Lucia Buck Foundation Awardee of the Life Sciences Research Foundation. B.T.G. and C.J.E.M. acknowledge support from Princeton Catalysis and Princeton Precision Health.

## Author contributions

E.H., T.C.W., and B.T.G. conceived of the study. E.H. performed the analysis and wrote the initial draft. All authors (E.H., T.C.W., J.S.C., S.D., R.G., K.M., C.J.E.M., S.W.P., C.V., and B.T.G.) reviewed and edited the manuscript.

## Competing interests

The authors declare no competing interests.
