## [Peer Review file · Nature Communications]

Using COVID-19 pandemic perturbation to model RSV-hMPV interactions and potential implications under RSV interventions

Corresponding Author: Dr Emily Howerton

Version 0:

Reviewer comments:

Reviewer #1

(Remarks to the Author)

In this study, the authors leveraged a mechanistic transmission model to explain variation in coupled RSV-hMPV outbreak patterns. They used post-pandemic rebound outbreaks as an out-of-sample test for their model. The major findings are that they found evidence for moderate interactions between RSV and hMPV, where RSV infection temporarily reduces the probability of hMPV transmission at the population level. Looking forward at the possible impacts of new RSV immunization strategies (monoclonal antibodies for infants and maternal vaccination to protect infants), their model did not appear to predict large hMPV outbreaks in the face of these new RSV interventions. Overall, this is a thought-provoking analysis with considerable novelty and sound methodology. The manuscript is very clearly written and the findings thoughtfully discussed. The major limitation of the study is that the model did not take age into account. Nevertheless, I think that it will be of significant interest to the readership, including public health practitioners, vaccine policy makers, an especially researchers in the field infectious diseases epidemiology and modelling. Furthermore, I believe in the importance of work such as this that objectively tries to assess potential unintended consequences of large scale public health interventions.

MAJOR COMMENTS

- As the authors point out, transmission and severity of RSV and hMPV are strongly age dependent. Age-specific patterns are not included in the model. This could affect the results in important ways, especially considering that the interventions available to prevent RSV in children are available only for a very specific age group: infants in their first months of life. Can the authors elaborate on how they might expect the lack of age data to influence their findings?
- I am not certain that I agree with the authors' conclusions that "surges of hMPV are improbable in the face of RSV interventions", that "the model does not predict a substantial risk of large hMPV outbreaks" or that findings suggest a "modest competitive release of hMPV". Figure S7 shows that the annual and peak numbers of hMPV cases could double, which would be substantial. Can the authors provide the estimates of the likelihood of a 50% increase or a doubling in annual hMPV cases due to interventions to prevent RSV?

MINOR COMMENTS

- Although it is true that childhood influenza vaccination was associated with increased rates of non-influenza viral infections in the study by Cowling BJ, et al (Clin Infect Dis 2012;54(12):1778-1783), these findings have not generally not been replicated (Sundaram ME et al, Clin Infect Dis 2013; 57(6):789-793).

(Remarks on code availability)

Reviewer #2

(Remarks to the Author)

This manuscript investigates the interactions between Respiratory Syncytial Virus (RSV) and Human Metapneumovirus (hMPV) by integrating epidemiological data with a mathematical modeling framework. The authors developed a two-pathogen transmission model and identified moderate, temporary effects of RSV infection on the transmissibility of hMPV. They assert that their model more accurately reproduces post-pandemic rebounds of RSV and hMPV. Moreover, the study

suggests that hMPV surges are unlikely to occur following RSV interventions, including active and passive immunization, likely due to the lower estimated transmissibility of hMPV compared to RSV. Overall, this work is highly relevant as it addresses critical questions about the interactions between paramyxoviruses and their implications for outbreak dynamics. It offers valuable insights into how these pathogens influence each other and highlights the potential impact of RSV vaccination programs on public health.

While the manuscript provides valuable insights, there are several areas where its presentation, methodology, and discussion could benefit from clarification or expansion. Below are detailed suggestions for improvement:

1. **Abstract:** The present format of Abstract is informative, but it is very dense. I believe it could benefit from a clearer structure to highlight key findings and implications succinctly. More comprehensive abstract is helpful for a wide audience, with more emphasis on public health implications of the findings.

2. **Data:** Statistical analysis needs to be more rigorous. Techniques like cross-wavelet transformation analysis could be employed to better understand phase differences and directional relationships between RSV and hMPV time series. Simple time-series plots are insufficient to capture nuanced relationships; additional quantitative analysis is necessary.

3. **Model and Model assumptions:**

(a) The assumptions underlying the SEIRS model, particularly the decision to focus on short-term interactions and exclude age-structured dynamics, need more justification. We know the transmission intensity and susceptibility are highly varied over different age-structures. While the rationale is briefly mentioned, a more detailed discussion of these trade-offs would enhance transparency.

(b) Authors have mentioned six types of interactions in their study. A clearer and comprehensive description is required of how model formulation really correlates those six types of interactions.

(c) Authors have used data from different geographical regions. The parameterization of seasonal forcing and its role in reproducing geographic variation could be elaborated. Were alternate functional forms tested, and how sensitive are the results to these choices?

(d) Authors have used very specific types of model discretization, without any explanation why this specific discretization is required, and how it actually helps in the dynamics of model and simulations!

(e) Also not understood, what is the focus of the study, – to describe the interaction between RSV and hMPV, or its rebound effects due to COVID-19 non-pharmaceutical controls! It is not clear in the abstract.

(f) Authors have used Good mobility data to implement its impact in the force of infection of RSV and hMPV. Although the methodology was taken from other literature, they need to describe with more details its implementation in their study.

(g) While the scalar parameter β is a useful simplification, it may oversimplify complex within-host and ecological interactions. Further discussion of limitations here is warranted.

4. **Discussion and results:**

(a) The manuscript references contrasting findings from prior studies but does not sufficiently explore why these discrepancies arise (e.g., differences in data resolution or modeling approaches).

(b) The "worst-case" scenarios are informative but could be expanded to consider broader uncertainties, such as vaccine uptake and global variations in RSV and hMPV epidemiology.

(c) The limitations are acknowledged, but some critical issues are underexplored. For example: the exclusion of RSV A/B subtype interactions might limit the model's predictive power. The potential role of coinfections or other viral interactions (e.g., influenza) is not addressed.

Specific Comments

1. The title effectively communicates the study's focus but could emphasize the modeling component to attract readers from computational epidemiology.

2. The abstract should include a brief mention of key limitations and areas for future research.

3. The introduction is thorough but could be streamlined. For instance, some discussion of RSV and hMPV biology might be condensed to focus more on the study's unique contributions.

4. Some technical terms (e.g., "competitive release," "strain replacement") are introduced without sufficient context for general readers.

5. Provide a flowchart or schematic of the SEIRS model to make it more accessible to readers unfamiliar with mathematical modeling.

6. The parameter estimation process is described but would benefit from additional context about prior studies or literature supporting the chosen priors.

7. The results section is well-structured but could highlight the broader significance of geographic differences (e.g., what drives the weaker seasonal forcing in Korea?).

8. Include a more detailed exploration of the potential policy implications of hMPV burden increases, particularly in regions with limited healthcare capacity.

So, my overall recommendation: With a moderate revision, this manuscript could be a valuable contribution to the field of epidemiology. It effectively bridges theoretical modeling and public health application but requires further clarity and contextualization to maximize its impact.

(Remarks on code availability)

Reviewer #3

(Remarks to the Author)

Review for manuscript "Using COVID-19 pandemic perturbation to model RSV-hMPV interactions and potential implications in an era of RSV interventions"

The article deals with an important topic. The manuscript is interesting and provides insight into the topic. The authors mentioned the limitations of their work which is excellent scientific practice.

The main aims of the article are difficult to grasp, and the writing can be improved. For instance, highlighting aims and results. The description of the methods needs to be improved. It seems that one main aim of the study is to study interactions between RSV and hMPV. To achieve this a mathematical model is proposed and fitted to data. Thus, the main results are related to the assumptions of the model and the fitting.

The following questions/suggestions can help to improve the paper. Some aspects must be addressed carefully.

Post-pandemic from 2018 to 2024? Please explain in detail.

Page 6. It is crucial that authors emphasize to the readers that hMPV transmission rate is forced by observed RSV incidence, $(1 + c)DRSV$. I think the word "can" should be changed to is. Also, deeper explanations to the sentence "as including both interactions provided too many degrees of freedom for model parameters to be meaningfully fit.". For instance, it is assumed RSV affects hMPV but not the other way. This is crucial in this study.

Page 6. Why is discretization needed? Please explain it in the manuscript.

The estimation of parameters needs to be explained in more detail since the main results of the study depend on it. Shape parameters for instance. There are many details not included. . Nowadays this is important for reproducibility.

Page 6. Underreporting is an important factor in the estimation of parameters. It is assumed that cases scale linearly with testing. In Figure 2 (supplementary material) the testing data is approximated by one-year moving average. The data shows a very seasonal year trend and variations between months are very large and are quite different from the moving average (solid line). Why not use a weekly underreporting rate that would be more accurate and might affect significantly the estimation of parameters?

The crucial part of the modeling is the coupling by means of parameters Dr_{sv} , phases (p_r and p_m), r and c . Table 3 does not show values of c . The value of c seems to be negative despite the authors mentioning that there is no strong evidence (page 3). This becomes confusing through the study. A negative value of c decreases the transmission rate of hMPV and then the peak appears later. However, the phases also play a similar role to delay the peak of hMPV. Thus, this can create issues with the identifiability of the model. This needs to be addressed carefully.

Table 2 shows a typo or confusing value for the reporting rate of hMPV. Please add comments about the lower reporting rate of hMPV.

Figure 4 needs to be commented on in detail. It seems that the model predicts similar trends of infected every year. The fit varies over time due to the reporting rate I suppose. This needs much better explanation. Readers would have many issues understanding the results and the methods.

From Table 2 there are many parameters that were estimated (more than 10). I don't think the model can be identified with the available data. Can you provide the identifiability analysis? Probably the practical one is the most important here.

Please add some clear comments about why someone would choose the coupled model over the independent one? Based on assumptions or due to a better fit to the data? Errors?

(Remarks on code availability)

Version 1:

Reviewer comments:

Reviewer #1

(Remarks to the Author)

I commend the authors for the quality of the additional/sensitivity analyses produced and of the clarifications provided. They have adequately addressed the issues that I raised

(Remarks on code availability)

Reviewer #2

(Remarks to the Author)

(Remarks on code availability)

Authors have considered all points raised by me and other reviewers, and revised the draft. Now, I recommend for acceptance of the paper.

Reviewer #3

(Remarks to the Author)

The article has been improved and the response letter addresses almost every point. I was surprised that in the revised version the changes are not highlighted to help the reviewers.

There are 14 non-informative priors in the model. The authors acknowledge this weakness but stated "that the post-pandemic rebounds provide a robust out-of-sample test for our parameter estimates and model fits. We can further test our model's ability to recreate outbreak patterns observed in other locations that were not used in model fitting."

I think the authors need to be more careful here and if a parameter identifiability is not possible (?) then make a weaker conclusion regarding the model fits and the consequences of this. The authors mentioned that they can try to use the model in other locations, but I don't think that would provide evidence of parameters' identifiability. In fact, there could be infinitely many combinations of parameter values that can provide a similar fit.

I think the Figure 5 is not mentioned in the text of the paper so readers will need to read the caption of it. I think this figure has many results that are not completely explained in the caption. I suggest to add full explanations on the main text. For example, Fig 5A seems to show the same likelihood for the independent and coupled models for the post-pandemic part. This is confusing since it seems going against the main conclusion of the paper. Similar aspects of the paper would be difficult to understand for many readers.

As I mentioned initially the article deals with an important topic. The manuscript is interesting and provides insight into the topic. The authors mentioned the limitations of their work which is excellent scientific practice.

(Remarks on code availability)

Response to reviews for *Using COVID-19 pandemic perturbation to model RSV-hMPV interactions and potential implications in an era of RSV interventions*

Reviewer #1

In this study, the authors leveraged a mechanistic transmission model to explain variation in coupled RSV-hMPV outbreak patterns. They used post-pandemic rebound outbreaks as an out-of-sample test for their model. The major findings are that they found evidence for moderate interactions between RSV and hMPV, where RSV infection temporarily reduces the probability of hMPV transmission at the population level. Looking forward at the possible impacts of new RSV immunization strategies (monoclonal antibodies for infants and maternal vaccination to protect infants), their model did not appear to predict large hMPV outbreaks in the face of these new RSV interventions. Overall, this is a thought-provoking analysis with considerable novelty and sound methodology. The manuscript is very clearly written and the findings thoughtfully discussed. The major limitation of the study is that the model did not take age into account. Nevertheless, I think that it will be of significant interest to the readership, including public health practitioners, vaccine policy makers, and especially researchers in the field infectious diseases epidemiology and modelling. Furthermore, I believe in the importance of work such as this that objectively tries to assess potential unintended consequences of large scale public health interventions.

Author response: Thank you for your comments.

MAJOR COMMENTS

- As the authors point out, transmission and severity of RSV and hMPV are strongly age dependent. Age-specific patterns are not included in the model. This could affect the results in important ways, especially considering that the interventions available to prevent RSV in children are available only for a very specific age group: infants in their first months of life. Can the authors elaborate on how they might expect the lack of age data to influence their findings?

Author response: We agree completely that age is an essential component of the transmission dynamics for these two pathogens, and thus we have added an age-structured version of our model to the supplementary material. The age-structured model provides equivalent population-level dynamics, but allows us to examine the age-specific infection probabilities, and how they change with and without RSV interaction. Changes include:

- We have added a paragraph at the end of the Methods section to describe this model, and provided a detailed description of the age-structured model in Supplementary Information Section 2.1.
- We have highlighted results from the age-structured model in a sentence added to the second paragraph of Results subsection *Implications for hMPV burden and dynamics under RSV interventions*: “To approximate changes in mean age of infection under the worst case scenario, we also incorporated heterogeneous, age-structured mixing into our fitted model (Supplementary Information Section 2). This suggested that eliminating the suppressive effect of RSV infection on hMPV transmission would decrease the mean age of hMPV infection (e.g., by approximately 3 weeks in children under three years of age, Supplementary Figure 10).”

In this analysis, we retain the blunt “worst-case” scenario that ignores age-specific intervention targets because age-specific interventions would restrict that proportion of the population for which hMPV is potentially released, making hMPV less likely to rebound in this scenario. However, you are correct that age-specific intervention targets will be important for making more precise predictions about the effects of RSV interventions. We believe a full exploration is warranted in a separate analysis. We have noted this limitation and the importance of future analyses in the seventh paragraph of the Discussion, which includes a reference to a recently published review of RSV pharmaceutical intervention development across age targets (Terstappen *et al.*).

- I am not certain that I agree with the authors’ conclusions that “surges of hMPV are improbable in the face of RSV interventions”, that “the model does not predict a substantial risk of large hMPV outbreaks” or that findings suggest a “modest competitive release of hMPV”. Figure S7 shows that the annual and peak numbers of hMPV cases could double, which would be substantial. Can the authors provide the estimates of the likelihood of a 50% increase or a doubling in annual hMPV cases due to interventions to prevent RSV?

Author response: Thank you for noting this point. Our original statements were intended to be made relative to pre-intervention RSV burden. However, as you point out, this was unclear as written. As such, we have made the following changes:

1. We have added Supplementary Figure 8 which shows the posterior probability of an x% increase in annual hMPV burden and annual hMPV peak magnitude. We also now report these results directly in the last paragraph of the Results section: “There is estimated to be a 20% probability that hMPV peaks double; yet, the model predicts that hMPV outbreak peaks will not exceed pre-intervention RSV peaks (Supplementary Figure 8).”
2. Throughout the text, we have edited statements that contain the claims you have flagged, which now read, respectively:
 - Abstract: “Finally, our model suggests that hMPV peak timing and magnitude may change in the face of RSV interventions, including active and passive immunization.”
 - First paragraph of the Discussion: “In a worst case scenario where new RSV interventions eliminate negative effects on hMPV transmission, the model predicts that increases in the peak size of hMPV outbreaks and changes in timing are possible. Yet, these changes are not predicted to exceed pre-intervention RSV burden, and thus overall combined burden is likely to decrease. ”
 - Fifth paragraph of Discussion: We have removed the claim that “surges in reported hMPV infections were unlikely” and now focus on hMPV burden relative to RSV: “Even under our conservative, “worst case” assumption, overall hMPV burden and peak incidence is predicted to be lower than pre-intervention RSV burden.”
 - Last paragraph of the Discussion: “Our model predicted that changes in peak timing and magnitude of hMPV outbreaks is possible in the face of RSV interventions; however, hMPV burden is not predicted to exceed pre-intervention RSV burden, likely because hMPV was estimated to be less transmissible than RSV.”

MINOR COMMENTS

- Although it is true that childhood influenza vaccination was associated with increased rates of non-influenza viral infections in the study by Cowling BJ, et al (Clin Infect Dis 2012;54(12):1778-1783), these findings have not generally not been replicated (Sundaram ME et al, Clin Infect Dis 2013; 57(6):789-793).

Author response: Thank you for bringing this reference to our attention. To avoid confusion, we have removed this example from the introduction.

Reviewer #2

This manuscript investigates the interactions between Respiratory Syncytial Virus (RSV) and Human Metapneumovirus (hMPV) by integrating epidemiological data with a mathematical modeling framework. The authors developed a two-pathogen transmission model and identified moderate, temporary effects of RSV infection on the transmissibility of hMPV. They assert that their model more accurately reproduces post-pandemic rebounds of RSV and hMPV. Moreover, the study suggests that hMPV surges are unlikely to occur following RSV interventions, including active and passive immunization, likely due to the lower estimated transmissibility of hMPV compared to RSV. Overall, this work is highly relevant as it addresses critical questions about the interactions between paramyxoviruses and their implications for outbreak dynamics. It offers valuable insights into how these pathogens influence each other and highlights the potential impact of RSV vaccination programs on public health.

Author response: Thank you for your comments.

While the manuscript provides valuable insights, there are several areas where its presentation, methodology, and discussion could benefit from clarification or expansion. Below are detailed suggestions for improvement:

1. Abstract: The present format of Abstract is informative, but it is very dense. I believe it could benefit from a clearer structure to highlight key findings and implications succinctly. More comprehensive abstract is helpful for a wide audience, with more emphasis on public health implications of the findings.

Author response: Thank you for the suggestion. We have added the following as the third sentence in the abstract to provide an overview of the study for readers before presenting the results: “Here, we use a mathematical model to quantify the likelihood of such interactions from population-level surveillance data and investigate the extent to which such interactions could lead to increases in hMPV burden in the face of RSV interventions.”

2. Data: Statistical analysis needs to be more rigorous. Techniques like cross-wavelet transformation analysis could be employed to better understand phase differences and directional relationships between RSV and hMPV time series. Simple time-series plots are insufficient to capture nuanced relationships; additional quantitative analysis is necessary.

Author response: This is a very good point. Because we are interested in the lags between RSV and hMPV outbreaks, we originally chose to focus on the time domain through cross correlation analyses. However, we agree that additional analyses in the frequency domain would further support our claims. We have added cross wavelet transform analyses for all locations to the Supplementary Material (now Supplementary Figure 2). We have updated subsection *Variation in RSV and hMPV outbreak patterns across multiple locations* to reference the cross wavelet transform analysis: “Across these locations, RSV and hMPV exhibit annual outbreaks where hMPV outbreaks consistently lag RSV outbreaks (Supplementary Figure 2).”

Unfortunately, the historical data available for Canada is not long enough to statistically validate our claims that British Columbia and the Prairies region exhibit major/minor biennial outbreaks before the pandemic; we have added caveats to these claims in Results subsection *Variation in RSV and hMPV outbreak patterns across multiple locations*. We have also removed the biennial labels in Figure 1 and put this caveat in the caption.

3. Model and Model assumptions:

- (a) The assumptions underlying the SEIRS model, particularly the decision to focus on short-term interactions and exclude age-structured dynamics, need more justification. We know the transmission intensity and susceptibility are highly varied over different

age-structures. While the rationale is briefly mentioned, a more detailed discussion of these trade-offs would enhance transparency.

Author response: We agree that these are important features of the analysis that were not well addressed. In response, we have performed sensitivity analyses on our results through two model extensions:

- i. Short-term interactions: Although there is evidence to suggest short-term interactions are likely at play and evidence for longer-term interactions is mixed (as described in the Introduction), you are correct that the duration of interaction between RSV and hMPV is largely unknown. We have added an extension to our model that allows us to begin to probe this. In particular, we allow lagged RSV incidence (e.g., summed over the past 6 weeks) to force the hMPV transmission rate. Using this framework, we re-estimate model parameters (including the interaction parameter) for lags of 0-7 weeks. Unfortunately, longer lags cannot be easily considered because the interpretation of the interaction term becomes complicated by reinfections that would be double counted. We have added a paragraph to the end of Results subsection *Population-level model to characterize presence of potential RSV-hMPV interactions* to discuss the results of this model extension.
- ii. Age structure: We agree completely that this is an essential component of the transmission dynamics for these two pathogens, and thus we believe a full exploration of these dynamics is warranted in a separate analysis. However, as a first step, we have added an age-structured version of our fitted model to the Supplementary Information. The age-structured model provides equivalent population-level dynamics, but allows us to examine age-specific infection probabilities and how these infection probabilities change with and without RSV interaction. We have highlighted these results in a sentence added to the second paragraph of Results subsection *Implications for hMPV burden and dynamics under RSV interventions*: “To approximate changes in mean age of infection under the worst case scenario, we also incorporated heterogeneous, age-structured mixing into our fitted model (Supplementary Information Section 2). This suggested that eliminating the suppressive effect of RSV infection on hMPV transmission would decrease the mean age of hMPV infection (e.g., by approximately 3 weeks in children under three years of age, Supplementary Figure 10).”

Below are additional changes to the Methods, Discussion, and Supplementary Information relating to the model extensions.

- i. Methods/Supplementary Information: We have added an additional paragraph to the Methods section *Out of sample simulations, sensitivity analysis, and model extensions* (formerly entitled *Post-pandemic rebounds and sensitivity analysis*) that describes these model extensions and their motivation. A detailed description of each model extension is also provided in Supplementary Information Section 2. There are 5 new supplementary figures (10-14) with results from these analyses that are referenced in the main text.
 - ii. Discussion: In the sixth and seventh paragraphs of the Discussion, we have also included both of these points as limitations to the current model and emphasized importance of future analyses in these areas.
- (b) Authors have mentioned six types of interactions in their study. A clearer and comprehensive description is required of how model formulation really correlates those six types of interactions.

Author response: Thank you for the suggestion. We agree that a more explicit link between the mechanisms we have presented and the formulation of our model is warranted. As such, we have made the following changes:

- i. We have restructured the beginning of the Results section *Population-level model to characterize presence of a potential RSV-hMPV interactions* to describe how the model parameters correspond to the mechanisms described in Figure 2. This includes explicitly linking the mechanisms of lower transmission rates and different seasonality to the no interaction hypothesis, and moving the paragraph that describes interaction mechanisms to where the interaction term, c , is introduced. We also provide an explicit interpretation of c : “We take c not significantly different from zero to imply no evidence for an interaction (i.e., no effect in Figure 2), and $c < 0$ to imply RSV infection reduces hMPV transmission rate (i.e., RSV infection makes individuals less susceptible to hMPV infection or makes future hMPV infection less transmissible”
 - ii. We have added a paragraph to the end of this section that discusses our model extension examining the duration of RSV-hMPV interaction. Importantly, in this paragraph, we now introduce the “per-capita” interpretation of the interaction parameter, which we believe will help readers directly relate our parameter estimates to biological mechanisms.
 - iii. In the subsection title and throughout, we have switched from referring to characterizing the “strength” of potential interactions to “presence” as we think this is more reflective of our model’s purpose.
- (c) Authors have used data from different geographical regions. The parameterization of seasonal forcing and its role in reproducing geographic variation could be elaborated. Were alternate functional forms tested, and how sensitive are the results to these choices?
 Author response: Other functional forms for seasonal forcing are possible in this type of model; however, most require the estimation of additional parameters. To keep the number of parameters to a minimum, we opted for a sine-wave seasonal forcing which is commonly used for respiratory viruses in temperate regions, such as Scotland. In addition, the majority of the data comes from a single region of Scotland (Glasgow), decreasing the likelihood that there are major variations in seasonality underlying the observed outbreaks. We have noted this important reasoning in Methods section *Transmission model*: “In both cases, we use a sine wave to capture seasonal changes in the transmission rate, which is commonly used for respiratory viruses in temperate regions such as Scotland. In addition, we use a single seasonal forcing function for the entire population as the majority of the data comes from Glasgow, decreasing the likelihood that there are major variations in seasonality underlying the observed outbreaks.”
- (d) Authors have used very specific types of model discretization, without any explanation why this specific discretization is required, and how it actually helps in the dynamics of model and simulations!
 Author response: This is a good point. We have added a new paragraph to the end of subsection *Transmission model* that outlines the motivation and rationale behind the chosen discretization scheme:
 “We use a discretized version of the above model for computational efficiency in parameter estimation. In particular, we discretized the model following He *et al.*, which assumes the number of events that occur follows a Poisson process. For a Poisson process occurring at rate λ , the probability of no events occurring is $e^{-\lambda\Delta t}$, and the probability of at least one event is $(1 - e^{-\lambda\Delta t})$. Thus, the number of individuals that leave compartment X in time step Δt is $\Delta X[t] = (1 - e^{-r\Delta t})X[t-1]$ assuming r is the sum of rates out of X in the ordinary differential equation model. This discretization scheme ensures that the number of individuals in each compartment will always be positive. See Supplementary Information for additional details and the full discretized model.”
- (e) Also not understood, what is the focus of the study, – to describe the interaction between RSV and hMPV, or its rebound effects due to COVID-19 non-pharmaceutical controls! It is not clear in the abstract.

Author response: Thank you for this question. The focus of our study was to describe and better understand possible interactions between RSV and hMPV using historical time series data. We fit a model with and without interactions, and we used the coupled post-pandemic rebounds of both viruses as an out of sample test. We have clarified this in the abstract: “We use post-pandemic rebounds of RSV and hMPV as an out of sample test for our model, and the model with interactions better predicts this period than a model where the pathogens are assumed to be independent.”

- (f) Authors have used Good mobility data to implement its impact in the force of infection of RSV and hMPV. Although the methodology was taken from other literature, they need to describe with more details its implementation in their study.

Author response: We have made edits to the second paragraph of Methods subsection *Out of sample simulations and sensitivity analyses* to better describe what the Google mobility data measures and clarify our process for including it in the model. This includes, among other minor edits, adding a sentence that describes the information provided in the Google mobility data: “The reports provide a single value for each week in a given location that represents the percent change to the number of visits to locations in each category compared to pre-pandemic baselines.”

- (g) While the scalar parameter c is a useful simplification, it may oversimplify complex within-host and ecological interactions. Further discussion of limitations here is warranted.

Author response: We agree this is an important point to emphasize. We have rewritten our discussion of these points to include more nuance (beginning of sixth paragraph in the Discussion):

“Important factors that could affect the interaction between RSV and hMPV may be omitted in our simple model. We used a single scalar term, c to phenomenologically capture all interactions between RSV and hMPV. While this approach provides useful foundation for detecting possible interactions, it risks oversimplifying complex biology such as immunological mechanisms of protection and the changing intensity of protection over time. In addition, our framework cannot easily detect the duration of viral interactions. We showed that lagged 7 weeks of past RSV incidence yielded a 1-1 per-capita effect of a single RSV infection in decreasing the transmission rate of hMPV. Per-capita effects of less than one from longer-duration interactions are plausible, but difficult to interpret in this framework given how we model reinfections. Future modeling work should focus on quantifying the duration of interaction as a means to differentiate innate and adaptive mechanisms.”

4. Discussion and results:

- (a) The manuscript references contrasting findings from prior studies but does not sufficiently explore why these discrepancies arise (e.g., differences in data resolution or modeling approaches).

Author response: We now explicitly state our hypotheses as to why our results may contrast: “We hypothesize that these findings contrast with ours (Nickbakhsh *et al.*[1] even used a subset of the same Scottish data) because the correlational analyses used monthly incidence data, which could mask the short-term interactions we have modeled, although further investigation is warranted.”

- (b) The “worst-case” scenarios are informative but could be expanded to consider broader uncertainties, such as vaccine uptake and global variations in RSV and hMPV epidemiology.

Author response: We have designed our worst-case scenario from the perspective of hMPV burden. hMPV burden is expected to be highest if there is absolutely no suppressive effect of RSV on hMPV (e.g., through the complete elimination of RSV infections

or through the lack of any cross-protective immunity for those who received interventions). However, you are correct, that this is indeed not the worst case scenario for public health overall (e.g., low intervention uptake would presumably be much worse in terms of combined burden). We have made the following changes to make this clearer:

- i. We have rewritten how we define the worst-case scenario in the last paragraph of the Results: “For each of these dynamical regimes, we also simulated potential effects of the introduction of RSV interventions. Since the effects of RSV interventions on the RSV-hMPV interaction are uncertain, we chose to model a generic “worst case” scenario from the perspective of hMPV public health burden, where immunity induced by a given RSV intervention is assumed to have no protective effect against future hMPV infection and RSV intervention coverage is complete. In other words, the suppressive effect of RSV infection on the hMPV transmission rate has been completely eliminated (i.e., $c = 0$ versus the fitted value $c = -7.13$; dashed orange lines, Figure 6C).”
- ii. In the Discussion, we have added the following sentences to note the importance of coverage in our worst-case scenario assumptions and emphasize the importance of considering other scenarios for future policy implications.
 - “Moreover, our assumption implies perfect coverage of RSV interventions, yet realized coverage will be imperfect.”
 - “The global policy implications of such predictions should also be explored.”
- (c) The limitations are acknowledged, but some critical issues are underexplored. For example: the exclusion of RSV A/B subtype interactions might limit the model’s predictive power. The potential role of coinfections or other viral interactions (e.g., influenza) is not addressed.

Author response: These are good points. We have added them to the seventh paragraph of the Discussion where we include limitations: “We could not model the dynamics of RSV A and B separately, which are known to have complex dynamics that could affect model predictions, and we did not consider potential interactions with other pathogens such as influenza.”

Specific Comments

1. The title effectively communicates the study’s focus but could emphasize the modeling component to attract readers from computational epidemiology.

Author response: Thank you for the suggestion. We have opted to leave the title as is, in an effort to attract a broad audience beyond computational epidemiology. Nevertheless, we believe that it will be of significant interest to the readership, including public health practitioners, vaccine policy makers, and especially researchers in the field infectious diseases epidemiology and modeling.

2. The abstract should include a brief mention of key limitations and areas for future research.

Author response: We have added this sentence at the end of the Abstract: “The analysis we present here provides a useful foundation for detecting possible RSV-hMPV interactions at the population level, although such a model certainly oversimplifies important complexities about the mechanisms by which interactions occur.”

3. The introduction is thorough but could be streamlined. For instance, some discussion of RSV and hMPV biology might be condensed to focus more on the study’s unique contributions.

Author response: Thank you for this suggestion. We have streamlined the introduction to focus primarily on RSV-hMPV interactions and the potential implications of this interaction in the face of new RSV interventions. In particular, we have removed unnecessary clinical detail from the first paragraph, removed the classification of interaction mechanisms which

is discussed later (presented in Figure 2), removed an example of pathogen interactions from the fifth paragraph, and streamlined repetitive text in the last two paragraphs. Some key references on RSV interventions have been moved to the third paragraph of the Discussion.

4. Some technical terms (e.g., “competitive release,” “strain replacement”) are introduced without sufficient context for general readers.

Author response: Good point. We have updated this sentence to provide more context: “This phenomenon of changing dominant serotypes was coined “strain replacement” and has parallels with “competitive release” in ecology, where reducing the density of one species increases the growth rate of another species that is competing for the same resource. ”

5. Provide a flowchart or schematic of the SEIRS model to make it more accessible to readers unfamiliar with mathematical modeling.

Author response: Thank you for this suggestion. We have added a model diagram to the main text (now Figure 3), which includes a list of key model assumptions. We reference this figure when we introduce the model (first and third paragraphs of the Results subsection *Population-level model to characterize strength of potential RSV-hMPV interactions*).

6. The parameter estimation process is described but would benefit from additional context about prior studies or literature supporting the chosen priors.

Author response: Thank you for this suggestion. We have added a new column to Supplementary Table 1 that describes the motivation behind our choices. We have also added references and context to the Methods subsection *Parameter estimation* where these assumptions are discussed: “We fixed the mean incubation time, the mean recovery rate, and the mean waning rate, and we assumed a constant birth and death rate that was calculated from demographic data to ensure a stable population size.”

7. The results section is well-structured but could highlight the broader significance of geographic differences (e.g., what drives the weaker seasonal forcing in Korea?).

Author response: We agree that explaining geographic differences across locations could provide additional evidence about the mechanisms of interaction between RSV and hMPV (e.g., differentiating ecological mechanisms such as host behavior change that may vary across locations and immunological mechanisms which are more likely to be consistent across locations). Although we are able to recreate the qualitative dynamics observed across regions by altering seasonal forcing in the model (as shown in the bifurcation diagram, Figure 5), we believe a separate analysis is necessary to fully explore the possible drivers of differences in patterns across locations. We have modified the second to last paragraph of the Discussion to further emphasize this important point.

8. Include a more detailed exploration of the potential policy implications of hMPV burden increases, particularly in regions with limited healthcare capacity.

Author response: Thank you for this suggestion. Because our “worst case” scenario suggests that hMPV burden will not exceed RSV burden in the pre-intervention period, we hypothesize that the overall burden on healthcare systems (i.e., RSV burden + hMPV burden) will decrease with RSV interventions. We believe a more detailed understanding of the mechanisms of interaction will enlighten exactly how RSV interventions are expected to affect hMPV and enable more detailed predictions and policy recommendations; thus, such an exploration would be better suited for a subsequent analysis. We have noted this at the end of the fifth paragraph of the Discussion.

So, my overall recommendation: With a moderate revision, this manuscript could be a valuable contribution to the field of epidemiology. It effectively bridges theoretical modeling and public health application but requires further clarity and contextualization to maximize its impact.

Author response: Thank you for thoughtfully commenting on our manuscript.

Reviewer #3

The article deals with an important topic. The manuscript is interesting and provides insight into the topic. The authors mentioned the limitations of their work which is excellent scientific practice. The main aims of the article are difficult to grasp, and the writing can be improved. For instance, highlighting aims and results. The description of the methods needs to be improved. It seems that one main aim of the study is to study interactions between RSV and hMPV. To achieve this a mathematical model is proposed and fitted to data. Thus, the main results are related to the assumptions of the model and the fitting.

Author response: Thank you for your comments.

The following questions/suggestions can help to improve the paper. Some aspects must be addressed carefully.

- Post-pandemic from 2018 to 2024? Please explain in detail.

Author response: Thank you for the clarifying question. We include one pre-pandemic season in our out of sample set to test the model on steady-state dynamics at the annual attractor in addition to the pandemic and post-pandemic periods. We have clarified this throughout the text:

- In the fourth paragraph of Results subsection *Population-level model to characterize presence of potential RSV-hMPV interactions*, we have added text to clarify this point: “To test the fitted model out of sample, we simulated outbreaks from July 2018 through March 2024. This period was chosen to include one season before as well as the period during and after the COVID-19 pandemic; we assumed that COVID-19 NPIs reduced RSV and hMPV transmission rates proportional to observed reductions in mobility.”
 - In Methods subsection *Parameter Estimation*, we have added “ $C_i[t]$ represents weekly counts of RSV and hMPV detections in Scotland, and we defined the fitting period as all weeks before June 16, 2018 in order to include one pre-pandemic season in our out of sample period.”
 - In the final subsection of the Methods *Out of sample simulations, sensitivity analysis, and model extensions*, we have clarified the purpose of these out of sample simulations: “(a) test a model’s ability to capture observed dynamics out of sample, including one pre-pandemic season as well as post-pandemic rebounds, and ...”
 - We have also added this information to the caption of Figure 4: “... The out of sample period was chosen to include one full pre-pandemic season (beginning in 2018), as well as all seasons during and after the COVID-19 pandemic perturbation. Performance is summarized across the full out of sample period, as well as for the pre- and post pandemic portions of the out of sample period, and for both pathogens (RSV in green, hMPV in orange). The pre-pandemic portion of the out of sample period is defined from June 25, 2018 - March 1, 2020, and the post-pandemic portion of the out of sample period is defined from January 1, 2021 - March 4, 2024 ...”
- Page 6. It is crucial that authors emphasize to the readers that hMPV transmission rate is forced by observed RSV incidence, $(1+c\text{DRSV})$. I think the word “can” should be changed to is. Also, deeper explanations to the sentence “as including both interactions provided too many degrees of freedom for model parameters to be meaningfully fit.”. For instance, it is assumed RSV affects hMPV but not the other way. This is crucial in this study.

Author response: As you suggest, c is the essential term in our model. First, we use the word “can” intentionally because we do not want to presuppose there is an effect of RSV on hMPV. If c is estimated to be zero, we conclude there is no effect of RSV on the transmission of hMPV. We have clarified this in the third paragraph of Results subsection *Population-level model to characterize presence of potential RSV-hMPV interactions*: “We take c not

significantly different from zero to imply no evidence for an interaction (i.e., no effect in Figure 2), and $c < 0$ to imply RSV infection reduces hMPV transmission rate (i.e., RSV infection makes individuals less susceptible to hMPV infection or makes future hMPV infection less transmissible).”

In addition, we have made the following changes to justify our assumption that RSV affects hMVP, but not vice versa:

- We now mention prior evidence that suggests RSV-hMPV interactions are asymmetric in the third paragraph of the introduction: “Although both negative and positive mechanisms are theoretically possible [2], RSV-hMPV interactions are generally hypothesized to be negative (i.e., hampering transmission or disease severity) and asymmetric (i.e., RSV affects hMPV more strongly than vice versa). ... In this study, RSV replication rate was unchanged by the presence of an hMPV coinfection or infection 2 days prior.”
 - When we introduce the model in the third paragraph of Results subsection *Population-level model to characterize presence of potential RSV-hMPV interactions*, we justify this assumption: “Our assumption of asymmetric interactions is based on evidence from prior modeling and experimental studies.”
 - In the sentence you highlight from the Methods section, we now focus solely on prior modeling and experimental evidence to support our assumption, as we do not provide a full analysis to justify our prior degrees of freedom claim: “We consider only the effect of RSV on hMPV given the results in Bhattacharyya *et al.* and Geiser *et al.* suggesting the effect of RSV on hMPV is stronger than vice versa.”
- Page 6. Why is discretization needed? Please explain it in the manuscript. The estimation of parameters needs to be explained in more detail since the main results of the study depend on it. Shape parameters for instance. There are many details not included. . Nowadays this is important for reproducibility.

Author response: Thank you for raising this question. We have added a new paragraph on discretization, and additional details about our choices of fixed parameters and priors. Changes include:

- We have added a new column to Supplementary Table 1 that provides justification for chosen parameters and priors, as well as references and context to the Methods subsection where these assumptions are discussed:
 - * “We fixed the mean incubation time, the mean recovery rate, and the mean waning rate, and we assumed a constant birth and death rate that was calculated from demographic data to ensure a stable population size.”
 - * “To account for potential overdispersion in the disease transmission process, we modeled the observed number of detections for each pathogen as $C_i[t] \sim \text{NegBinom}(\rho_i \tau [t] I_i[t], \phi_i)$ for $i \in \{\text{RSV}, \text{hMPV}\}$ and some shape parameter ϕ_i which is estimated based on the dispersion of the data”.
- We have added a new paragraph at the end of subsection *Transmission model* to motivate and describe the discretization scheme in more detail: “We use a discretized version of the above model for computational efficiency in parameter estimation...”
- We have made edits to the second paragraph of Methods subsection *Out of sample simulations, sensitivity analyses, and model extensions* to better describe what the Google mobility data measures and clarify our process for including it in the model.
- We have noted the reasoning behind our choice of seasonal forcing function in Methods section *Transmission model*: “In both cases, we use a sine wave to capture seasonal changes in the transmission rate, which is commonly used for respiratory viruses in temperate regions such as Scotland. In addition, we use a single seasonal forcing function

for the entire population as the majority of the data comes from Glasgow, decreasing the likelihood that there are major variations in seasonality underlying the observed outbreaks.”

- Page 6. Underreporting is an important factor in the estimation of parameters. It is assumed that cases scale linearly with testing. In Figure 2 (supplementary material) the testing data is approximated by one-year moving average. The data shows a very seasonal year trend and variations between months are very large and are quite different from the moving average (solid line). Why not use a weekly underreporting rate that would be more accurate and might affect significantly the estimation of parameters?

Author response: This is a good point. We have performed a sensitivity analysis where we incorporated weekly trends in testing in addition to the annual trends we originally used (Supplementary Figure 10). Estimated parameters and model dynamics are qualitatively similar to the model without considering weekly testing. We have added a reference to this figure in the first paragraph of the *Parameter estimation* subsection where we introduce our assumptions about testing: “... see Supplementary Figure 10 for additional analysis where incorporation of weekly testing trends yielded qualitatively similar model fits and dynamics”.

- The crucial part of the modeling is the coupling by means of parameters Dr_{sv} , phases (pr and pm), r and c . Table 3 does not show values of c . The value of c seems to be negative despite the authors mentioning that there is no strong evidence (page 3). This becomes confusing through the study. A negative value of c decreases the transmission rate of hMPV and then the peak appears later. However, the phases also play a similar role to delay the peak of hMPV. Thus, this can create issues with the identifiability of the model. This needs to be addressed carefully.

Author response: You are correct, we use c to test potential interactions between RSV and hMPV, and it is very important to distinguish this from the phase of seasonal forcing. In the coupled model where RSV can affect the transmission of hMPV through the term c , the phase of the seasonal forcing function is assumed to be the same for both pathogens. Thus, if the phase shifts to accommodate hMPV seasonal forcing, the model will not fit RSV dynamics well. However, independent dynamics with different seasonal forcing is the null hypothesis. Thus, we compare the performance of the coupled model to an independent model where the phase of RSV and hMPV differ. We show that this model performs equally well in the pre-pandemic period, but not as well in the post-pandemic period (Figure 4, fourth paragraph of *Population-level model to characterize presence of potential RSV-hMPV interactions*). To clarify these points, we have made the following changes:

- We have reframed c to be interpreted as a per-capita rate. We have added an additional paragraph that discusses this interpretation of c (final paragraph of Results subsection *Population-level model to characterize presence of potential RSV-hMPV interactions*). Corresponding minor changes have been made to the presentation of c in the Methods (Equation 6 and related text). The reformulated equation is equivalent to the old version, but highlights the per-capita interpretation of c .
- We have restructured the first few paragraphs of the Results section *Population-level model to characterize presence of potential RSV-hMPV interactions* where the model is introduced. This includes providing a detailed the interpretation of c (“We take c not significantly different from zero to imply no evidence for an interaction (i.e., no effect in Figure 2), and $c < 0$ to imply RSV infection reduces hMPV transmission rate (i.e., RSV infection makes individuals less susceptible to hMPV infection or makes future hMPV infection less transmissible).”).
- Throughout the text, we have changed “negative effect” to “suppressive effect” to be more clear in the interpretation of c .

- We have added a new figure (now Figure 3) that includes a diagram of our model. The modeling assumptions are clearly listed.
 - You were correct, we had inadvertently omitted c from Supplementary Table 2. We have fixed this, so that c is now included.
- Table 2 shows a typo or confusing value for the reporting rate of hMPV. Please add comments about the lower reporting rate of hMPV.

Author response: Thank you for noting this, we have fixed the typo in Supplementary Table 2. In addition, we have added a note to the fifth paragraph of the Discussion that the hMPV reporting rate may be unrealistically low: “In addition, the model predicted lower reporting of hMPV infections compared to RSV, although the estimate of hMPV reporting rate may be unrealistically low.”

- Figure 4 needs to be commented on in detail. It seems that the model predicts similar trends of infected every year. The fit varies over time due to the reporting rate I suppose. This needs much better explanation. Readers would have many issues understanding the results and the methods.

Author response: We agree, this point needed additional clarification. We have updated the fourth paragraph of the Results subsection *Population-level model to characterize strength of potential RSV-hMPV interactions* as follows: “Much of the year-to-year variation in detections of RSV and hMPV outbreak magnitude before the pandemic were explained by variations in testing, which we accounted for in the model with data on the number of tests that were performed (Methods, Supplementary Figure 3). The number of tests performed decreased after the pandemic, yet post-pandemic outbreaks were predicted to be larger due to a buildup of susceptible individuals during the pandemic NPI period (number of susceptible and infected individuals, Figure 4).”

- From Table 2 there are many parameters that were estimated (more than 10). I don’t think the model can be identified with the available data. Can you provide the identifiability analysis? Probably the practical one is the most important here.

Author response: Parameter identifiability in epidemiological models is known to be a challenge, and recent studies have emphasized the importance of the type of data used to make inference[3, 4]. We fixed the biological parameters that could be obtained from the literature to account for these issues. Moreover, we believe the post-pandemic rebounds provide a robust out-of-sample test for our parameter estimates and model fits. We can further test our model’s ability to recreate outbreak patterns observed in other locations that were not used in model fitting. Nonetheless, this is an important point to note, and we have now included in the seventh paragraph of the Discussion: “To overcome known identifiability issues with the epidemiological models including the SEIR model, we fixed biological parameters that could be obtained from the literature. Although this could affect our conclusions, we believe the post-pandemic rebounds provide a robust out-of-sample test for our model fits.”.

- Please add some clear comments about why someone would choose the coupled model over the independent one? Based on assumptions or due to a better fit to the data? Errors?

Author response: Thank you for this suggestion. We have added the following sentences to the second paragraph of the Discussion to highlight this important feature of our results: “Our phenomenological model, which assumed hMPV transmission dynamics were coupled to RSV incidence, better captured post-pandemic rebound dynamics than the null model that assumed independent RSV and hMPV outbreak dynamics. The post-pandemic period was essential to distinguishing between these two models, as pre-pandemic patterns could be captured in the independent model with different seasonal forcing of the transmission rate for each pathogen.”

Response to reviews for *Using COVID-19 pandemic perturbation to model RSV-hMPV interactions and potential implications under RSV interventions*

June 30, 2025

Reviewer #1

I commend the authors for the quality of the additional/sensitivity analyses produced and of the clarifications provided. They have adequately addressed the issues that I raised

Author response: Thank you for your suggestions to improve our manuscript.

Reviewer #2

Authors have considered all points raised by me and other reviewers, and revised the draft. Now, I recommend for acceptance of the paper.

Author response: Thank you for your suggestions to improve our manuscript.

Reviewer #3

The article has been improved and the response letter addresses almost every point. I was surprised that in the revised version the changes are not highlighted to help the reviewers.

Author response: Thank you for your suggestions to improve our manuscript.

There are 14 non-informative priors in the model. The authors acknowledge this weakness but stated “that the post-pandemic rebounds provide a robust out-of-sample test for our parameter estimates and model fits. We can further test our model’s ability to recreate outbreak patterns observed in other locations that were not used in model fitting.” I think the authors need to be more careful here and if a parameter identifiability is not possible (?) then make a weaker conclusion regarding the model fits and the consequences of this. The authors mentioned that they can try to use the model in other locations, but I don’t think that would provide evidence of parameters’ identifiability. In fact, there could be infinitely many combinations of parameter values that can provide a similar fit.

Author response: We agree with the reviewer that parameter identifiability is a concern. Our primary test of the model is reproducing post-pandemic rebound outbreaks out of sample. However, as you note, it is not a perfect test. For example, post-pandemic dynamics depend on assumptions about how NPIs affected transmission rates during this period, which are uncertain. Thus, we have performed an additional sensitivity analysis to further increase the rigor of our analysis and added additional caveats throughout. Both are described below.

In the main text, we assumed that NPIs affected RSV and hMPV transmission rates equally, but, this may not necessarily be the case (e.g., if different age groups are driving transmission). In an additional sensitivity analysis (now presented in Supplementary Figure 16), we have relaxed this assumption. We still find evidence that the independent model does not reproduce post-pandemic outbreaks well compared to the coupled model. In addition to comparing the coupled and independent models, we also tested the performance of an intermediate model: the coupled model, which assumes the same seasonality for both, with the interaction parameter set to zero.

We find that the coupled model with interaction can seemingly reproduce post-pandemic dynamics, as you suggest, but this is because the model overestimates out of sample pre-pandemic outbreaks. Thus, we believe the coupled model still provides the best fit to explain both the pre-pandemic outbreaks and the transient dynamics after the pandemic.

Despite our attempts to make this out-of-sample test as robust as possible, we acknowledge that parameter identifiability is still a concern. Thus, we have reworded our conclusion in the Abstract and removed the word “robust” to describe the post-pandemic rebound as an out of sample test. We have also created a separate Discussion paragraph devoted to parameter identifiability concerns by restructuring ideas that were already present and adding an additional sentence to caveat our results: “Nevertheless, this test depends on assumptions about NPIs which are highly uncertain, and it is still possible that alternative parameter sets could fit pre- and post-pandemic dynamics equally well.” Finally, we have added a reference to Supplementary Figure 16 in the Results and Methods sections.

I think the Figure 5 is not mentioned in the text of the paper so readers will need to read the caption of it. I think this figure has many results that are not completely explained in the caption. I suggest to add full explanations on the main text. For example, Fig 5A seems to show the same likelihood for the independent and coupled models for the post-pandemic part. This is confusing since it seems going against the main conclusion of the paper. Similar aspects of the paper would be difficult to understand for many readers.

Author response: We discuss the results of Figure 5 in the sixth paragraph of section *Population-level model to characterize presence of potential RSV-hMPV interactions*. The second to last sentence of this paragraph states that the results for post-pandemic hMPV outbreaks were not significantly different when using log-likelihood: “When using log likelihood to assess performance, there was no significant difference between post-pandemic predictions for hMPV.”

As I mentioned initially the article deals with an important topic. The manuscript is interesting and provides insight into the topic. The authors mentioned the limitations of their work which is excellent scientific practice.

Author response: Thank you again for carefully considering our manuscript and suggesting improvements.